# Diel patterns in stream nitrate concentration produced by in-stream processes

Jan Greiwe[1], Markus Weiler[1], Jens Lange[1]

[1]Hydrology, University of Freiburg, Friedrichstraße 39, 79098 Freiburg, Germany

*Correspondence to*: Jan Greiwe (jan.greiwe@hydrology.uni-freiburg.de)

**Abstract.** Diel variability in stream $NO_3^-$ concentration represents the sum of all processes affecting $NO_3^-$ concentration along the flow path. Being able to partition diel $NO_3^-$ signals into portions related to different biochemical processes would allow
calculation of daily rates of such processes that would be useful for water quality predictions. In this study, we aimed to identify distinct diel patterns in high-frequency $NO_3^-$ monitoring data and investigated the origin of these patterns. Monitoring was performed at three locations in a 5.1 km long stream reach draining a 430 km² catchment. Monitoring resulted in 355 complete daily recordings on which we performed a k-means cluster analysis. We compared travel time estimates to time lags between monitoring sites to differentiate between in-stream and transport control on diel $NO_3^-$ patterns. We found that travel time failed
to explain the observed lags and concluded that in-stream processes prevailed in the creation of diel variability. Results from the cluster analysis showed that at least 70% of all diel patterns reflected shapes typically associated with photoautotrophic $NO_3^-$ assimilation. The remaining patterns suggested that other processes (e.g. nitrification,denitrification and heterotrophic assimilation) contributed to the formation of diel $NO_3^-$ patterns. Seasonal trends in diel patterns suggest that the relative importances of the contributing processes varied throughout the year. These findings highlight the potential in high-frequency
water quality monitoring data for a better understanding of the seasonally in biochemical processes.

## 1    Introduction

In-stream processing of nutrients can significantly influence loads and concentrations transported to receiving ecosystems (Peterson et al., 2001; Roberts and Mulholland, 2007). Nutrients are repeatedly taken up and released again by organisms during downstream transport, a concept known as "nutrient spiraling" (Ensign and Doyle, 2006). Depending on the rates of
nutrient uptake and release, in-stream nutrient processing may reduce the risk of harmful eutrophication (Birgand et al., 2007).

Among the different nutrients, nitrate ($NO_3^-$) is of special interest since it usually represents the largest fraction in dissolved inorganic nitrogen and is nowadays easily detectable using in-situ optical spectrometer probes. At the same time, water quality management requires knowledge of $NO_3^-$ processing rates to predict how rapidly $NO_3^-$ inputs are transformed and attenuated. This is particularly relevant in light of a changing climate and a predicted reduction of summer flow (Austin and Strauss, 2011; Mosley, 2015; Hellwig et al., 2017).

Similar to other solutes, e.g. dissolved oxygen (DO) or carbon dioxide ($CO_2$), $NO_3^-$ concentrations can exhibit diel (i.e. 24 h) patterns. However, the increasing body of high frequency $NO_3^-$ monitoring data from optical in-situ probes shows that such diel patterns are not ubiquitous. Some streams consistently exhibit strong diel patterns (Heffernan and Cohen, 2010), while others do so only during certain seasons (Rusjan and Mikoš, 2010; Aubert and Breuer, 2016; Schwab, 2017; Rode et al., 2016), and still others do not show diel patterns at all (Duan et al., 2014). Biochemical processes influencing $NO_3^-$ concentration include $NO_3^-$ depletion via denitrification (Mulholland et al., 2009) and both autotrophic (Lupon et al., 2016b) and heterotrophic (Middelburg and Nieuwenhuize, 2000; Luque-Almagro et al., 2011) assimilation, as well as production via mineralization and subsequent nitrification. Previous studies have suggested that diel variation in stream $NO_3^-$ concentration are mainly related to in-stream photoautotrophic uptake (Nimick et al., 2011; Burns et al., 2019). Due to photosynthetic light requirements, photoautotrophs take up $NO_3^-$ mostly during the day (Mulholland et al., 2006), which causes minimum and maximum $NO_3^-$ concentrations to typically occur in the late afternoon and in the early morning (prior to sunrise), respectively. However, there is evidence that diel variation may not be influenced by photoautotrophic uptake alone. In many systems, diel variability has also been found in rates of nitrification and denitrification (Laursen and Seitzinger, 2004; Dunn et al., 2012; Scholefield et al., 2005), e.g. due to changing oxygen levels in sediments (Christensen et al., 1990).

In flowing waters, biochemical processes are superposed by downstream transport. Therefore, the solute signal measured at a specific location integrates over all conditions and events that water parcels were previously exposed to. As a result, the benthic footprint, i.e. the upstream area influencing concentrations at the measurement point, depends on flow velocity and solute turnover rate. While gaseous solutes like DO may quickly equilibrate with the atmosphere, upstream discontinuities in non-gaseous solutes like $NO_3^-$ (e.g. tributary confluxes, lakes or reservoirs, waste water inputs from point sources) may persist further downstream (Hensley and Cohen, 2016). In open systems with unknown input signals, it is therefore unclear whether diel concentration patterns are produced by conditions in the investigated stream or stem from upstream sources from which they are transported downstream (Pellerin et al., 2009). We will refer to the first case as in-stream control and to the second case as transport control. In this sense, we use the term 'in-stream' to refer to average properties of a reasonably long stream reach and its immediate surroundings including biochemical conditions in the stream and in the hyporheic zone as well as diffuse groundwater interaction.

In the present study, we analyze high-frequency $NO_3^-$ data observed at three monitoring sites delimiting two reaches in the lower course of the river Elz in Southwest Germany. We aim to investigate, (1) if there are diel patterns in $NO_3^-$ concentration, (2) if these patterns are subject to in-stream or transport control, and (3) how they are related to environmental conditions and potential drivers. In order to address these questions, we performed a cluster analysis on $NO_3^-$ recordings. We differentiated

between in-stream and transport control by comparing travel time estimates to time lags between concentration signals at adjacent monitoring sites. Finally, we compared environmental conditions among clusters and determined correlations between the concentration rates of change and potential drivers of biochemical processes.

## 2 Methods

### 2.1 Study site

The studied stream reach is located in the lower course of the river Elz in Southwest Germany between the municipalities of Emmendingen and Riegel (Figure 1). At our study site, the river Elz drains an area of approximately 430 km² of which 66% are forest and 21% are grassland. The fraction of cropland is below 2%. The river contains several weirs and receives inflows from a small wastewater treatment plant approximately 25 km upstream of the study site. Yet, most wastewater of the upstream catchment is transferred to a large treatment plant located further downstream. The monitored stream section spans a distance

of 5.1 km and is divided into two reaches with different morphology. The upper reach (2.7 km) is characterized by a uniform gravel bed, which is straightened and protected against erosion by regularly spaced groundsills. The lower reach (2.4 km) was subject to extensive revitalization including flood dam relocation and installation of a near-natural meandering river course. Revitalization measures were completed in 2016 and since then natural dynamics have controlled river morphology. Both reaches are characterized by largely open canopies and shallow (usually below 0.4 m) water depths, which allows light to reach

the stream bed. However, in the downstream reach water depths are more variable, exceeding 1.5 m at some locations. As a consequence, also flow velocities are more variable in the downstream reach. Both reaches are scarcely colonized by macrophytes and filamentous algae. A visible biofilm was observed on the gravel bed, particularly in the second half of the growing season. There are no obvious influxes along the two stream reaches, except for a minor tributary in the upstream reach (Fig. 1).


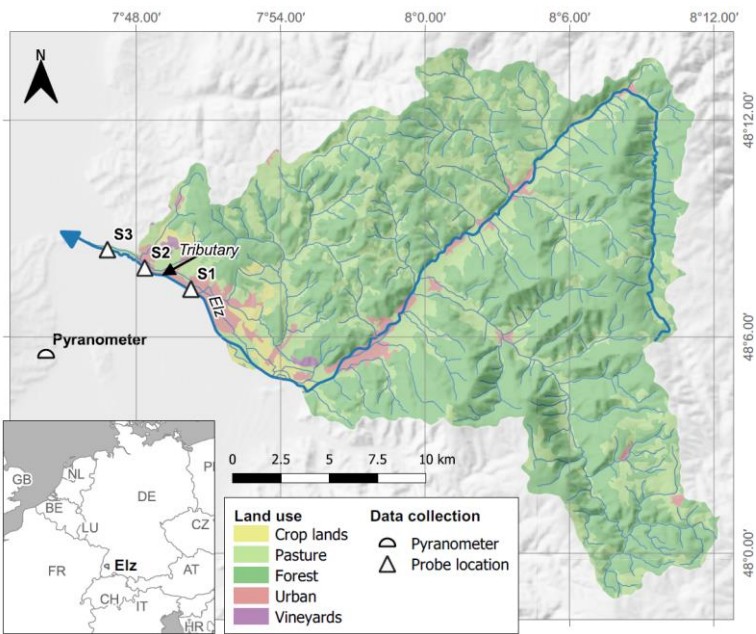

**Figure 1: Location of monitoring points along the stream reach and land use in the contributing catchment.**

## 2.2    Data collection

Concentration of $NO_3^-$ was measured at 15 min intervals at the three monitoring sites (S1, S2, S3) using UV-Vis spectrometer
probes (spectro::lyser, s::can Messtechnik GmbH, Vienna, Austria) from April to November in 2019. As only two spectrometer
probes were available, one probe was periodically repositioned so that input and output concentrations of either the upper or
the lower stream reach were measured. In addition, biweekly grab samples were collected at eight locations along the studied
stream reach, including the probe locations, to provide a local calibration for probe measurements (Fig. S1) and to assess
longitudinal concentration evolution between monitoring sites. Samples were analyzed using ion chromatography (Dionex
ICS-1100, Thermo Fisher Scientific Inc., USA). Stream temperature (T) and water levels (h) were continuously recorded at
the downstream site S3 (TD-Diver, Van Essen Instruments, Netherlands) at 15 minute intervals. Discharge was calculated
using a rating-curve based on two salt dilution measurements by the regional water authority, and one additional salt dilution
measurement during higher water levels on 15 November 2019 (> 70 % of recorded water levels). In the latter dilution
measurement, we injected 33 kg of NaCl at site S1 to cover both reaches. We used global irradiance (S) data from a nearby
climate station (< 10 km, Fig. 1) as a measure of sunlight intensity.

## 2.3 Data analysis

### 2.3.1 Identification of patterns in stream $NO_3^-$ concentration

Patterns in $NO_3^-$ concentration may also vary on the seasonal, daily, and the spatial scale. We assessed seasonal relationships of absolute $NO_3^-$ concentrations and diel $NO_3^-$ variability to environmental conditions by determining Spearman rank correlations of daily means of $C_{obs}$ and daily $NO_3^-$ amplitudes with global irradiance (S), water temperature (T), and, water level (h). We used k-means cluster analysis to identify diel patterns in stream $NO_3^-$ concentrations as done previously by Aubert and Breuer (2016). This method partitions a data set into a pre-defined number of k clusters by iteratively minimizing the within cluster sum of squares. We used the algorithm by Hartigan and Wong (1979) that is implemented in the 'stats' R-package (R Core Team, 2019). The input to this algorithm is a matrix whose rows represent elements to be partitioned (days in the present case) and whose columns represent the dimensions according to which the elements are compared. In the present case, these dimensions correspond to the time of day of the measurements (n=96 at a measurement interval of 15 minutes). More information about the method can be found in Tan et al. (2019).

The k-means analysis was done on the diel portion of the solute concentration signal, hereafter referred to as diel concentration ($C_{diel}$), to ensure that the resulting clusters represented variability in diel cycles and not in $NO_3^-$ background concentrations. Diel (residual) concentrations were obtained by subtracting a 24 hour centered moving average from the measured concentrations ($C_{obs}$) and smoothed by applying a moving average of 2 hours. One feature of the k-means method that introduces some degree of subjectivity is the determination of the number of clusters k. We therefore tested k values between 2 and 20 and determined the best partition by both an assessment of explanatory benefit per additional cluster, also known as 'elbow method', and by visual inspection of clusters. The elbow method was not clearly indicative as no sharp bent was observed. Instead, we visually found an optimum number of six clusters, since higher values of k did not produce new clusters in terms of timing but rather caused further splitting of existing clusters by amplitude. As a measure for longitudinal stability of diel patterns, we calculating the fraction of days on which diel patterns at the upstream and downstream monitoring sites of each reach were assigned to the same cluster.

### 2.3.2 In-stream vs. transport control on diel $NO_3^-$ patterns

In order to differentiate between in-stream and transport control on diel $NO_3^-$ patterns, we determined time lags between adjacent monitoring sites by cross-correlation analysis and compared these to estimated solute travel time. If diel $NO_3^-$ variation originated from some upstream source and subsequent downstream transport, time lags between sites should correspond to solute travel times. In contrast, if diel patterns were produced by in-stream processes more or less simultaneously at all points along the flow path, we expected the time lag to be zero in most instances. Cross-correlation analysis is a standard method to

determine time lags between signals (Derrick and Thomas, 2004). It is based on the idea that the strength of a correlation between two signals changes according to a temporal shift. The shift that maximizes the strength of the correlation is considered the time lag between the signals. This method works best, if the two signals have a similar shape, i.e. they are strongly correlated at an optimal lag. We therefore determined time lags only for days when the correlation coefficient r between upstream and downstream sites exceeded 0.75. This was true for 29 out of 42 completely recorded days in the upper and 92 out of 121 complete days at the lower reach.

Time lags were compared to two independent estimates of solute travel time: mean tracer travel time ($\tau_a$) and nominal water residence time ($\tau_n$) according to Kadlec (1994). While $\tau_a$ is the first moment of the tracer residence time distribution and was determined from the breakthrough curves of our salt dilution measurements, $\tau_n$ is the ratio of reach volume and discharge. In contrast to $\tau_a$, which requires tracer data as an input and could only be determined for our own dilution measurement (raw data of low flow measurements was not available from the regional water authority), $\tau_n$ was calculated continuously from water level recordings and channel width. As discharge, water depth, and channel width vary along the stream reach, we decided to account for variability in channel geometry and flow conditions by estimating a range of likely travel times based on channel width. Channel widths were estimated from aerial imagery and ranged from 20 to 25 m in the lower sub-reach and from 15 to 20 m in the upper sub-reach. Time lags obtained from cross-correlation were tested for difference from zero using t-tests and for difference from travel time estimates using paired t-tests.

### 2.3.3 Characterization of clusters

In order to characterize the clusters, we compared environmental parameters during the occurrence of the respective clusters. Particularly, we assessed daily means of $NO_3^-$ concentration, water levels ($h_{mean}$), and water temperature ($T_{mean}$) as well as the daily maximum solar irradiance ($S_{max}$). Differences between clusters were statistically assessed using analysis of variance (ANOVA) and Tukey honestly significant difference (HSD) tests as implemented in the 'stats' R-package (R Core Team, 2019).

The relationships between clusters and potential drivers were investigated by calculating daily Spearman rank correlations between $C_{diel}$ and the diel course of the drivers. As potential drivers we considered global irradiance (S), water temperature (T) and discharge, the latter represented by water level (h). These environmental parameters are usually considered to influence the rate of biogeochemical processes, i.e. the rate of change of $NO_3^-$ concentration rather than instantaneous $NO_3^-$ concentration. Laboratory experiments have shown such behavior for the effect of light on $NO_3^-$ uptake rates of algae (Grant, 1967) or the effect of temperature on denitrification rate (Pfenning and McMahon, 1997). We therefore assessed correlations between drivers and the first derivative ($\delta C_{diel}$) of the diel concentration signal $C_{diel}$. This corresponds to the way biochemical processes

are implemented in some recent solute models (Hensley and Cohen, 2016; Grace et al., 2015). However, changes in water
level may affect $NO_3^-$ concentrations both indirectly, e.g. by influencing hyporheic exchange and biochemical processes therein
(Trauth and Fleckenstein, 2017), and directly, since additional flow components may be enriched or depleted in $NO_3^-$ compared
to pre-event water. In the case of water level, we therefore calculated correlations with both $C_{diel}$ and $\delta C_{diel}$.

## 3    Results

### 3.1    Variability of diel patterns in space and time

Data collection at the three monitoring sites resulted in 352 complete diel $NO_3^-$ recordings. Throughout the season, stream
$NO_3^-$ concentrations ranged between 2.47 mg $L^{-1}$ and 7.44 mg $L^{-1}$ (Fig. 3). Mean values and standard deviations at the three
monitoring sites were $4.64 \pm 0.66$ mg $L^{-1}$ (S1), $4.63 \pm 0.73$ mg $L^{-1}$ (S2), and $4.36 \pm 0.75$ mg $L^{-1}$ (S3). Considering only days
with complete upstream and downstream observations, i.e. comparing averages of the same day, $NO_3^-$ concentration
significantly increased between S1 and S2 (from 4.61 to 4.86 mg $L^{-1}$, p<0.001, n=42) and significantly decreased between S2
and S3 (from 4.54 to 4.40 mg $L^{-1}$, p<0.001, n=121) (Fig. S2). Daily averages of $NO_3^-$ were negatively correlated with water
level ($\rho$=-0.34, p<0.001), positively with water temperature ($\rho$=0.53, p<0.001), and uncorrelated with global irradiance
($\rho$=0.01, p=0.93). The overall negative correlation between $NO_3^-$ and water level was dominated by large floods in May and
June. Particularly in the second half of the study period, e.g., in early August (S2 and S3) and late October (S1 and S2), $NO_3^-$
concentrations increased in response to floods (Fig. 3). Daily $NO_3^-$ amplitudes were neither correlated with water level
($\rho$=-0.03, p=0.76), water temperature ($\rho$=0.11, p=0.22), nor with global irradiance ($\rho$=-0.07, p=0.21).

The cluster analysis resulted in 6 clusters that clearly differed in terms of amplitude and timing of minimum and maximum
concentrations (Fig. 2). 69.6% of the days were attributed to the clusters A (n=128) and B (n=119), which both reached peak
concentration in the early morning and minimum concentration in the late afternoon, but the daily amplitude was higher in
cluster B. The remaining clusters were characterized by peaks around midday (cluster C, n=48), in the afternoon (cluster D,
n=28) and around midnight (cluster E, n=26). The last cluster (cluster F, n=3) did not include enough days for a proper
characterization. Average time of the daily concentration maxima in clusters A to E were 4:33 h, 5:32 h, 10:18 h, 16:44 h, and
23:33 h, respectively. The respective average times of daily minima were 17:46 h, 17:36 h, 21:39 h, 6:06 h, and 11:58 h.

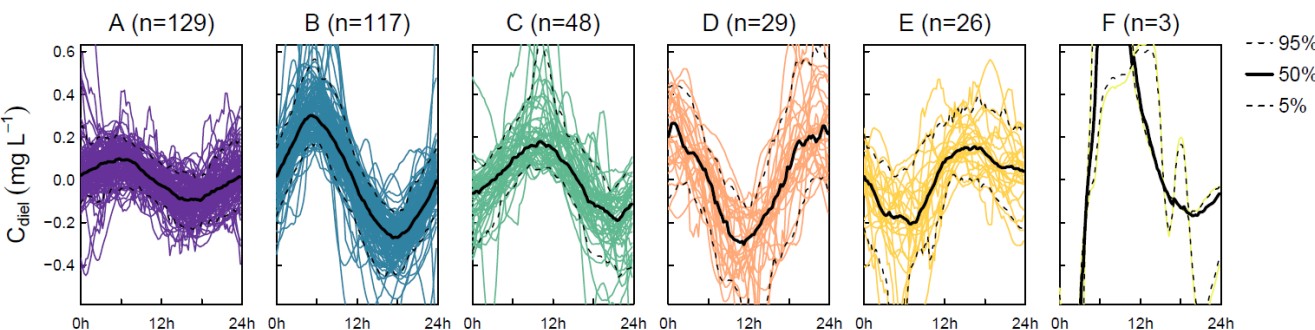

**Figure 2: Clusters found in diel residuals of NO₃⁻ concentration ($C_{diel}$). Capital letters above panels are cluster names ordered alphabetically according to cluster size. Black lines indicate median diel patterns, dashed lines indicate the 5th and 95th percentile. Note that $C_{diel}$ reflects deviations from the 24 h floating average so that negative values do not imply that negative concentration were observed.**

In terms of cluster occurrence, a largely similar seasonal pattern was apparent at all monitoring sites, despite different numbers of recorded days (Fig. 3). Cluster A dominated in May and again in October and was replaced by cluster B during the summer months from June to September. Both clusters usually formed continuous blocks of several days. Cluster C occurred occasionally throughout the season but preferentially in early summer, while cluster D and E mainly occurred in fall. On most days (62.0%), diel NO₃⁻ recordings at the upstream and downstream monitoring sites were attributed to the same cluster, i.e. these patterns were longitudinally stable. However, longitudinal stability differed between stream reaches (50.0% in the upper and 66.1% in the lower reach) and among clusters. Cluster A was most stable (84.2%, n=57), while cluster B (62.3%, n=53) and C (61.9%, n=21) were close to the average. Cluster D (28.6%, n=14) and cluster E (12.5%, n=16) turned out to be comparatively unstable.

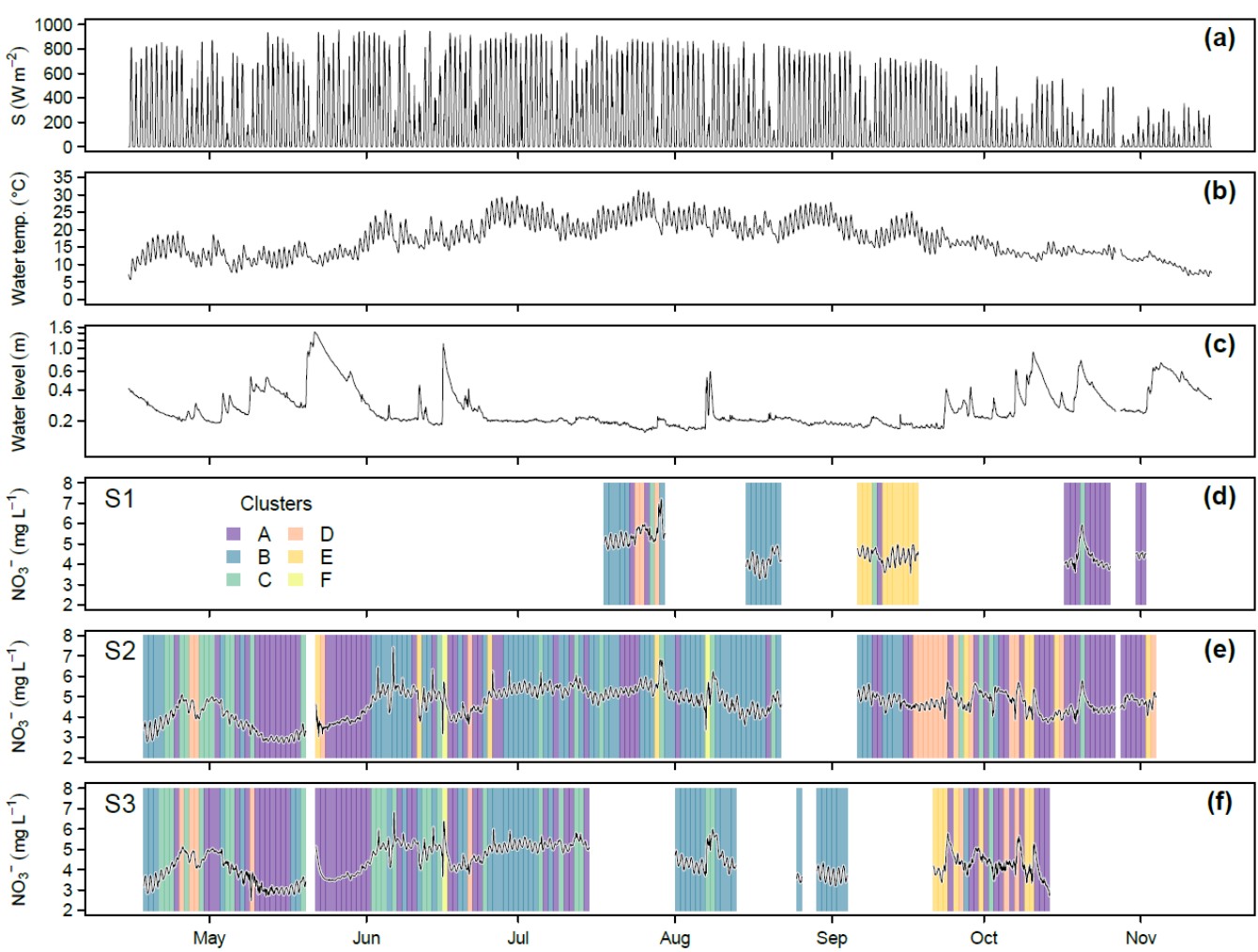

**Figure 3: Global irradiance (a), water temperature (b) and water level (c) at S3 as well as NO$_3^-$ concentration and cluster occurrence at the monitoring sites S1 (d), S2 (e), and S3 (f). Background colors in panels d to f indicate to which cluster the corresponding day was assigned.**

## 3.2 In-stream vs. transport control on diel patterns

The time lags between diel NO$_3^-$ signals at adjacent monitoring sites were usually shorter than the solute travel times between the stations. The salt dilution measurement resulted in a discharge of 2.0 m$^3$ s$^{-1}$ and travel time ($\tau_a$) estimates of 2.0 h in the upper and 2.3 h in the lower reach (Fig. 4). Estimates of nominal residence time ($\tau_n$) increased with decreasing stream flows. The fact that the independently determined $\tau_a$ was included in the range of $\tau_n$, showed that the estimated travel times were

plausible. In both reaches, the time lags between the concentration signals roughly ranged between zero and the travel time estimates, but were significantly different from both zero (p<0.001, both reaches) and minimum travel time (p<0.001, both reaches). In the lower reach, lags formed an evenly distributed point cloud. Within this cloud, Cluster D, E, and F only appear at above median flows. In the upper reach, time lags were concentrated towards the extremes, i.e. either close to zero or close to travel time estimates. Days with below median stream flow were mainly assigned to cluster B and those above median stream flow to cluster A.

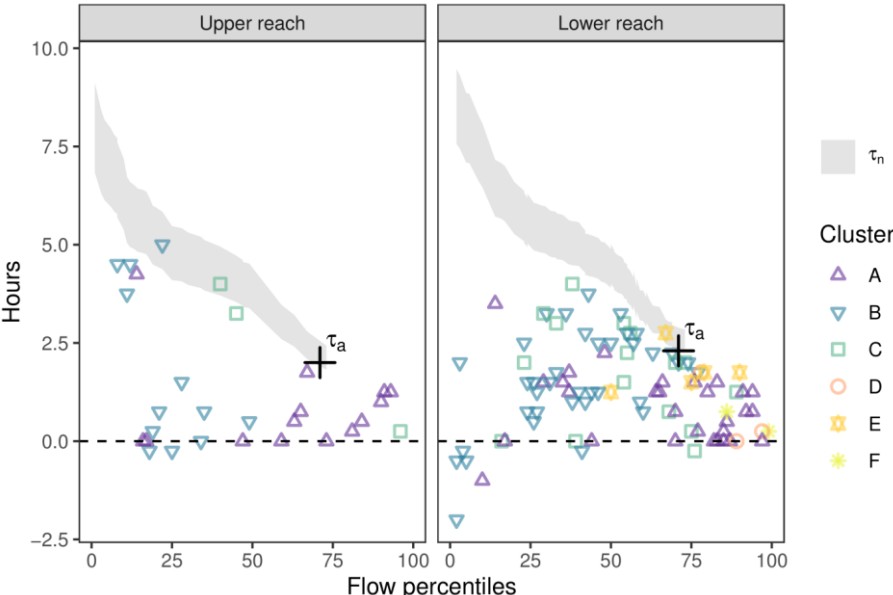

**Figure 4: Time lag between diel NO$_3^-$ signals at adjacent monitoring points (colored symbols) compared to the tracer travel time ($\tau_a$, black cross) and the range of nominal travel time estimates ($\tau_n$, shaded area). No travel times were estimated when discharge exceeded the validity range of the rating curve. The figure only shows lags determined from signals with a corresponding cross-correlation coefficient above 0.75 (84.0% of the days). Points falling below the shaded areas indicate in-stream control on diel NO$_3^-$ patterns, whereas points within the range of travel time estimates suggest transport control.**

### 3.3 Characterization of clusters

We found clear differences in the distribution of daily means of environmental parameters among clusters (Fig. 5). The following characterization of the clusters refers to significant differences (p<0.05) according to Tukey HSD test applied to an ANOVA on the cluster data. Cluster A presented overall lowest NO$_3^-$ concentrations (median 4.36 mg L$^{-1}$) which differed from those during cluster B (median 4.87 mg L$^{-1}$) and C (median 4.88 mg L$^{-1}$). Cluster A also showed the lowest water

temperature (median 14.1 °C) and elevated water levels (median 41.9 cm) compared to cluster B and C. Cluster B was

220 characterized by the highest global irradiance (median 825.0 W/m²), highest water temperature (21.7 °C) and lowest water levels (median 21.2 cm). Disregarding cluster F with only 3 data points, the difference in water temperature was significant for all remaining clusters. Global irradiance in cluster B differed from all clusters but C. Water level differed with all clusters but C and E. Cluster C occurred during very similar conditions as cluster B and only differed from cluster B in terms of water temperature (median 16.4 °C). The two clusters D and E were characterized by lower global irradiance than cluster B and C

225 and did not differ from one another. Cluster F consisted of only 3 days, but all of these represented water levels (median 77.2 cm) hardly ever observed in the remaining clusters.

In addition to different environmental conditions, we identified different relationships with potential drivers of diel cycles among clusters (Fig. 6). The correlation of $\delta C_{diel}$ and S was positive in cluster D, negative in clusters A and C, and strongly negative in cluster B. Moderate correlations of $\delta C_{diel}$ and T were found in cluster C (negative) and cluster E (positive).

230 Correlations of $\delta C_{diel}$ with h were weak and the differences among clusters were less pronounced than with S and T. The relationship of $C_{obs}$ and h was very variable and included both strongly positive and negative correlations.

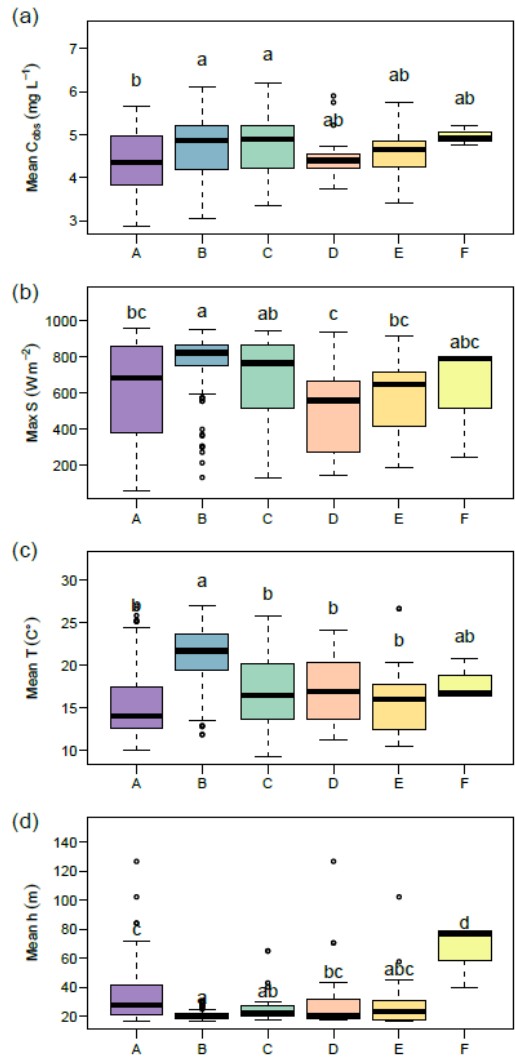

**Figure 5: Environmental conditions during occurrence of clusters.** The panels show daily average $NO_3^-$ concentration (a), daily maximum of global irradiance (b), daily average water temperature (c), and daily average water level (d). Lowercase letters above boxplots were assigned to groups that do not differ significantly according to analyses of variance (ANOVA) and Tukey tests.

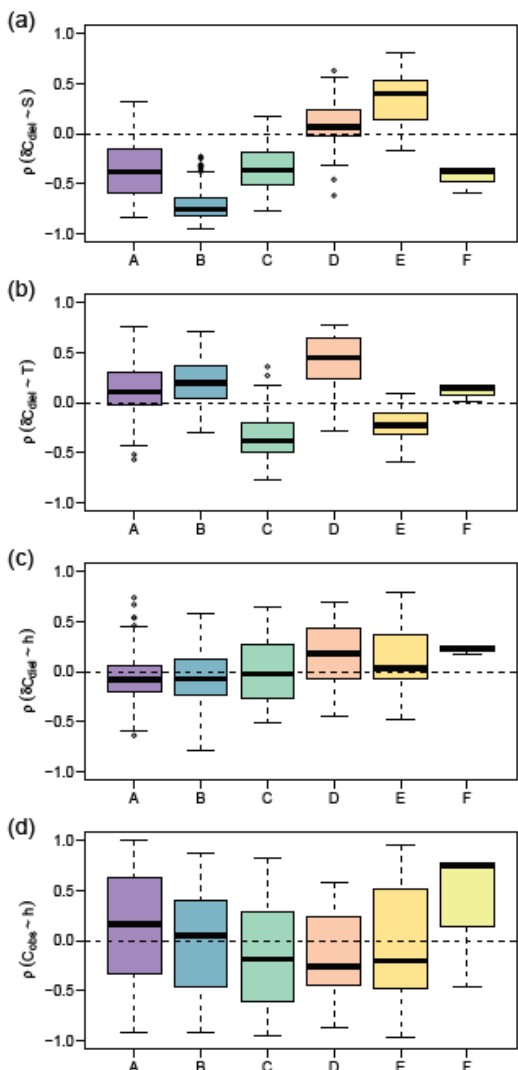

**Figure 6: Daily Spearman correlations of the NO₃⁻ signal with potential drivers by cluster.** The panels show correlation strength of diel concentration change rate with global irradiance (a), diel concentration change rate with water temperature (b), diel concentration change rate with water level (c), and observed concentration with water level (d).

# 4    Discussion

## 4.1    General patterns

In our data, we found patterns in $NO_3^-$ concentration both on the diel and on the seasonal scale. On the seasonal scale, a weak negative correlation of $NO_3^-$ and water level indicated that flow events tended to dilute $NO_3^-$ concentrations. However, particularly after the low flow period in summer, $NO_3^-$ increased during discharge events, an observation that is often explained by the mobilization of previously accumulated $NO_3^-$ in soils (Burns et al., 2019; Lange and Haensler, 2012). The fact that $NO_3^-$ was correlated with stream temperature but not with global irradiance may be a consequence of a more intense seasonal pattern in water temperature than in irradiance, since we started our monitoring campaign in late spring when daily irradiance peaks were already close to their seasonal maximum. On the diel scale, we identified six different $NO_3^-$ patterns that varied seasonally. Although daily $NO_3^-$ amplitudes would usually be expected to be highest on days with high metabolic activity (Rode et al., 2016; Heffernan and Cohen, 2010), daily amplitudes did not show correlations with daily averages of light intensity, water temperature or water level. The fact that longitudinal stability varied among cluster suggests that less stable clusters (e.g. D and E) either indicated a shift in in-stream conditions between monitoring sites or external controls on diel patterns, e.g. transport.

Variability in $NO_3^-$ concentration may also be influenced by lateral inputs, including tributaries and groundwater interaction. The only surface tributary within the studied stream reach was between S1 and S2. It was initially considered negligible and therefore not accounted for. However, snap shot sampling on a hot day during low flow conditions revealed nitrate concentration twice as high as in the main stream. Consequently, it is possible that the tributary influenced average $NO_3^-$ levels. It may also be subject to its own diel $NO_3^-$ fluctuations. Yet, the influence on patterns in the main stream must have been small as diel patterns were usually longitudinally stable, i.e. the same upstream (S1) and downstream (S2) of the tributary.

It is also possible that diffuse groundwater influx influenced average $NO_3^-$ concentration at the monitoring sites. In fact, $NO_3^-$ levels in regional groundwater wells were higher than in stream water in the proximity of the upper reach and lower than in stream water along the lower reach (Fig. S3). Although the overall flow direction of groundwater was parallel to the stream, groundwater inputs might explain the increase in average $NO_3^-$ concentration from S1 to S2 and subsequent decrease from S2 to S3 (Fig. S2). Previous research identified diffuse groundwater inputs as a considerable challenge for determining mass balances using paired high-frequency probes (Kunz et al., 2017). Due to these difficulties, we were unable to separate the effects of diffuse groundwater inputs from a potential effect of increased $NO_3^-$ removal in the lower reach due the revitalization measures.

## 4.2    Interpretation of diel patterns

The comparison of time lags between monitoring sites with travel time revealed that lags were usually too small to be produced by transport alone, but higher than expected in the case of pure in-stream control (Fig. 4). The presence of time lags may thus be caused by an interaction of transport and in-stream processes. Simulating the longitudinal evolution of $NO_3^-$ concentration downstream of a constant source, Hensley and Cohen (2016) found that timing of $NO_3^-$ extremes was variable in the proximity of the source. With increasing travel distance, however, $NO_3^-$ concentration converged into a stable signal solely defined by

in-stream processes. Depending on the position of observation points along such a stream reach, one may find time lags like those observed in our study. Although in our study boundary conditions were far less constrained than in the simulation of Hensley and Cohen (2016), their results might principally explain our observed time lags. Non-zero lags would then indicate that at the study site $NO_3^-$ concentration had not yet fully converged and was still partially influenced by transport. Nevertheless, observed time lags were clearly smaller than estimated travel times. We therefore conclude that the observed

diel $NO_3^-$ patterns were not primarily produced by transport.

In the following, we therefore aim to interpret our findings based on in-stream processes. In terms of biochemical processes, diel $NO_3^-$ variability depends on the time-varying balance of $NO_3^-$ removal (via assimilation by both heterotrophs and autotrophs as well as denitrification) and $NO_3^-$ production (via mineralization and subsequent nitrification). We do not regard our interpretations on the controls of the observed patterns complete but as hypotheses for further research on diel dolute

patterns to build upon. Considering the idea of multiple superposed biochemical processes as a starting point, some assumptions can be made on the diel course of the processes mentioned above. Photoautotrophic assimilation depends on the light availability and can be conceptualized as a function of solar irradiance. In contrast, the degree of diel variability in nitrification, denitrification, and heterotrophic assimilation is less clear. However, we generally assume that the rate of microbial metabolism (besides other influences) increases with temperature. Figure 7 schematically illustrates the diel

concentration signals resulting from overlaying the diel courses of light-dependent photoautotrophic ($r_{aa}$) assimilation and a complementary temperature-dependent processing rate ($r_{comp}$). The latter represents the combined net effect of heterotrophic assimilation, nitrification and denitrification. Particularly, we consider different levels of light intensity (columns in Fig. 7) and different types of relationship with temperature (rows in Fig. 7), including positive and negative correlation with temperature as well as constant $r_{comp}$. The shapes of $r_{aa}$ and $r_{comp}$ reflect means over all measurements of global irradiance and

water temperature, respectively, during the course of the present study.

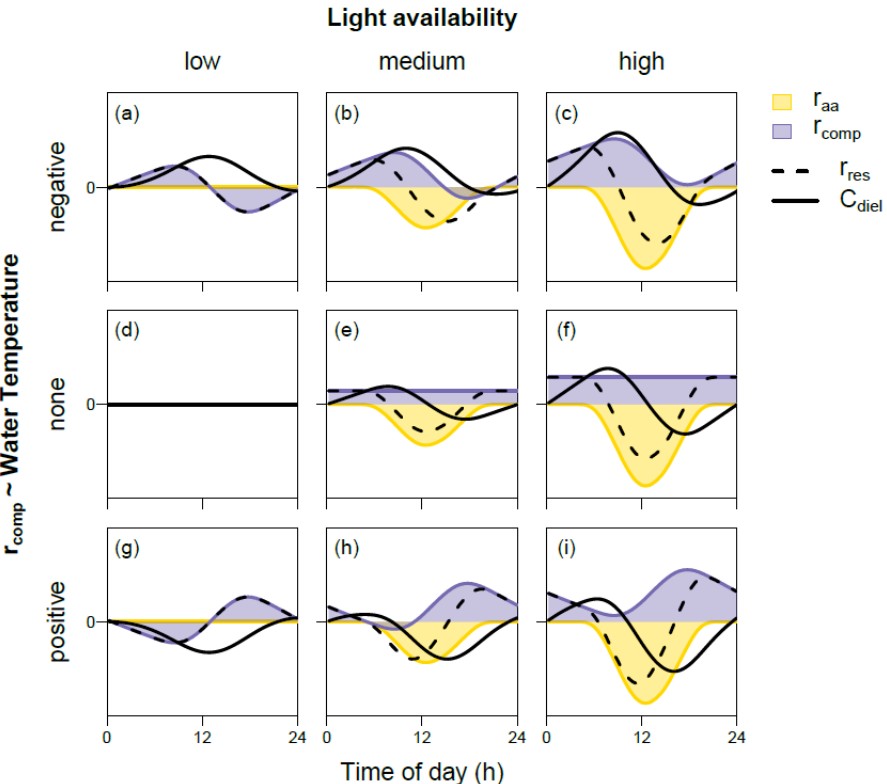

**Figure 7: Schematic representation of diel courses of assimilation rate ($r_{aa}$.) and a complementary processing rate ($r_{comp}$) required to produce equilibrium conditions. Black lines show the resulting change rate ($r_{res}$) and concentration (C). While $r_{aa}$ is considered a function of global irradiance (columns represent different levels of irradiation intensity), $r_{comp.}$ is conceptualized as a function of stream temperature (rows represent a negative (a-c), no relationship (d-f) and a positive relationship (g-i)).**

Diel $NO_3^-$ patterns with a maximum in the early morning and a minimum in the afternoon are usually explained by photoautotrophic $NO_3^-$ uptake by primary producers (Nimick et al., 2011; Heffernan and Cohen, 2010). This was also the largest group of diel patterns in our study, including cluster A and B, jointly accounting for about 70 % of the data. In our study, the idea that such diel patterns reflect photoautotrophic uptake is supported by a strongly (cluster B) and moderately (cluster A) negative correlation between $\delta C_{diel}$ and global irradiance. The higher amplitude of cluster B (Fig.2) may result from either stronger photoautotrophic $NO_3^-$ uptake compared to cluster A (Fig. 7e and f) or added effects of intense assimilation and diel variability in the interplay of other $NO_3^-$ depleting or producing processes (Fig. 7c and i). Consequently, the seasonality in cluster occurrence suggests that photoautotrophic $NO_3^-$ uptake was strongest from June to early September, when cluster B

prevailed. In May and October the dominance of cluster A suggests reduced photoautotrophic $NO_3^-$ uptake which may be due to reduced light availability in autumn or due to lower water temperatures and higher flow during both periods. The latter may have influenced photoautotrophic $NO_3^-$ uptake via reduced light penetration through a higher water layer, via disruption of stream metabolism due to destruction of vegetation by flood events (Burns et al., 2019).

Patterns with a midday maximum such as those observed in cluster C are hard to explain by photoautotrophic assimilation alone in systems without intense seasonal shading by riparian vegetation (as opposed to e.g. Rusjan and Mikoš (2010)). Figure 7 (a, b, c, f) shows that $NO_3^-$ peak time gradually shifts towards midday when photoautotrophic assimilation decreases and the relative importance of a $NO_3^-$ depleting process negatively related with temperature increases. This suggests that either denitrification or heterotrophic assimilation or both are promoted by stream temperature and drive the shape of the signal. Diel
variability has been observed both in denitrification (Christensen et al., 1990; Harrison et al., 2005; Cohen et al., 2012) and heterotrophic respiration (Hotchkiss and Hall, JR., 2014) which is closely linked to heterotrophic $NO_3^-$ assimilation. However, peak $NO_3^-$ depletion occurs in the afternoon, when oxygen levels are expected to be elevated and unfavorable for anaerobe denitrification (Rysgaard et al., 1994). In addition, it is not clear how denitrification in the lower anoxic sediments could be promoted by temperature without simultaneously increasing nitrification in the upper sediment layers. However, it seems
possible that the driving force of cluster C was not temperature but another process with similar diel pattern. Exudation of algal photosynthate rich in labile organic carbon (Kaplan and Bott, 1982) may have a similar diel course and stimulate assimilation and denitrification by heterotrophs but not nitrification by autotrophs. Under such conditions, heterotrophic assimilation or denitrification or both may drive diel $NO_3^-$ fluctuation in cluster C.

In literature, diel patterns with a $NO_3^-$ peak in the afternoon (cluster D) have been attributed to intense evapotranspiration
(Aubert and Breuer, 2016; Flewelling et al., 2014; Lupon et al., 2016a). In the present study, evapotranspiration was not measured, however, it did not produce systematic diel fluctuations in water level and the latter were not correlated with diel $NO_3^-$ signals. Such patterns could also not be reproduced by overlaying rates of light-dependent and temperature-dependent processes (Fig. 7). A satisfactory explanation for cluster D is therefore still needed.

Diel patterns with a midday low (cluster E) could be the result of low photoautotrophic assimilation and a temperature-
dependent $NO_3^-$ producing processes like nitrification (Fig. 7g). Diel variability in nitrification is well documented (Warwick, 1986; Laursen and Seitzinger, 2004; Dunn et al., 2012) and it seems principally plausible that temperature promotes nitrification without influencing denitrification in deeper anoxic sediment layers. Another reason for independence of nitrification and denitrification may be limitation of heterotrophic denitrification in absence of an organic carbon source.

These findings suggest that, despite a dominance of photoautotrophic assimilation, other processes contribute to the formation of diel $NO_3^-$ patterns in the river Elz. These may be adverse processes like nitrification on the one hand and denitrification and heterotrophic assimilation on the other hand. The relative importance of these processes varies seasonally and is reflected in shifts of diel $NO_3^-$ patterns. Although the distinct clusters identified in our analysis invite for speculation, in-stream $NO_3^-$ processing is complex and processes overlap and interact which makes unambiguous interpretation solely based on $NO_3^-$

recordings challenging.

## 4.3 Conclusions

In a 5.1 km stream reach of the river Elz in Southwest Germany, we identified diel patterns in stream $NO_3^-$ concentration, differentiated between in-stream and transport control, and analyzed how patterns were related to environmental conditions and potential drivers. We found a set of six clusters representing different characteristic diel $NO_3^-$ patterns. Relatively small

temporal shifts between adjacent monitoring sites indicated that $NO_3^-$ concentration patterns were predominantly formed by in-stream processes and not by a transport of upstream $NO_3^-$ inputs. Most patterns were characterized by a pre-dawn maximum and an afternoon minimum of varying intensity, and mostly the change rate of $NO_3^-$ concentration was negatively correlated with global irradiance. We therefore conclude that these patterns were primarily produced by photoautotrophic $NO_3^-$ uptake. However, we also found indications that other biochemical processes like nitrification and heterotrophic respiration contributed

to the formation of $NO_3^-$ patterns. In depth interpretation and eventually quantification of process rates would require spatially distributed high frequency information on stream metabolism, e.g. dissolved oxygen concentrations, and on different N species. Nevertheless, our analysis suggests that particular combinations of different in-stream processes may generate distinct diel $NO_3^-$ patterns. A seasonal shift in patterns may then indicate shifts in the relative importance of the underlying processes. The clustering method used in this study proved useful for making the data set accessible for this kind of analysis and may be used

as a blueprint for the analysis of other stream solutes.

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
