# Peer review of "Diel patterns in stream nitrate concentration produced by in-stream processes"

_Biogeosciences, 2020_

## Referee Comment (RC1) · Anonymous Referee #1 · 29 Dec 2020

Summary

This paper examines patterns and sources of diel variation in stream NO3 concentration along a lowland river in Germany. The authors show that diel patterns of stream NO3 concentration vary over the growing season, yet most days show similar diurnal oscillations. Further, by combining different statistical techniques, the authors convincingly show that diel patterns are mostly driven by in-stream processes. Finally, the authors analyze diel and seasonal patterns of several environmental variables to discuss which in-stream process is driving diel NO3 cycles.

General comments

This paper makes a significant contribution to watershed and stream ecology through its assessment of patterns and controls of diel variation in stream NO3 concentration. However, I have some major issues that need to be addressed. All comments are made in the spirit of increasing the potential impact of this interesting research.

1. While most of the findings presented in the paper are original and compelling, the conclusions raised from them are sometimes speculative and inaccurate. For instance, the authors concluded that "the magnitude of microbial NO3 processing may be large compared to plant uptake", but they did not measure any in-stream process (GPP, denitrification, nitrification) nor NO3 uptake rates. Hence, it is impossible to know, based on their data and results, which in-stream process was contributing the most to NO3 uptake rates over the study period. Similarly, they stated that "diel patterns in NO3 concentration suggest the importance of microbial pathways for in-stream processing", but the 70% of diel patterns seem to be driven by photoautotrophic uptake (not microbial pathways). My suggestion is to focus the objectives and conclusions on the compelling results and only speculate about the relative importance of different in-stream processes in the discussion.

2. I missed some results regarding lateral inputs. In the discussion, the authors mentioned that lateral inputs may not affect diel NO3 patterns because they did not observe diel variations in discharge. While I agree with this statement, lateral inputs should be included in the hypothesis, methods and results (see Flewelling et al. 2014 or Lupon et al. 2016). Also, the authors mentioned that there was a tributary entering to the upstream reach. Does the tributary show diel variation in NO3 concentration? How this may influence stream NO3 concentration in S2?

3. I was confused by some of the approaches used. For instance, what is the point of the mass balance? It has many uncertainties (e.g. groundwater, tributaries) and the results derived from it are difficult to interpret. My suggestion is to delete this whole section. Instead, I will focus on analyzing (i) if all sites showed similar seasonal patterns in diel NO3 variation (i) if the effect of longitudinal propagation differed across

clusters; (iii) if there was a lag time between diel patterns of drivers and stream NO3 concentration (see my specific comments for more info on this regard).

4. The discussion is a little bit puzzling. My suggestion is to delete all sub-headings and focus on how different sources shape stream NO3 concentration. You can start with a paragraph discarding longitudinal propagation and lateral inputs as factors causing diel NO3 patterns. Then, move to the most obvious process: photoautotrophic uptake (clusters A-B) and how it varies over time depending on light, temperature, discharge. Finally, you can suggest potential explanations for the other clusters: denitrification (cluster C), nitrification (cluster D), storm flow (cluster F).

5. While I like the figures, most of them (and their captions) need some improvements (see my specific comments). Also, I missed a figure showing the raw data (i.e. diel patterns of NO3, discharge, light and temperature over the whole study period). This figure is key to understand some of the points discussed (e.g. no diel variation in discharge); and it will be very helpful to the readers.

Specific comments

Ln 1. The title is a little bit speculative. Perhaps something focused on in-stream processes vs longitudinal propagation would be better.

Ln 21. This sentence is not accurate. What your results are telling us is that different in-stream processes might generate diel patterns in NO3 concentration, and that the relative importance of such processes may vary depending on the season.

Ln 37-44. This rationale is correct, but does not engage with the objective of the paper (i.e. you don't quantify any in-stream process). My suggestion is to delete this part and merge this paragraph with the following one.

Ln 46-55. As it occurs with the previous paragraph, this section goes beyond the objectives of the paper. My suggestion here is to shorten it to something like "Previous studies have suggested that seasonal patterns of diel variation in stream NO3 concentration are related to in-stream photoautotrophic uptake (refs). Due to photosynthetic light requirements, photoautotrophs take up NO3 mostly during the day, with minimum and maximum NO3 concentrations occurring at X and Y (refs). However, there is evidence that diel variation (. . .)."

Ln 67. What is the difference between the two hypothesis? They look exactly the same to me. Be explicit with the hypotheses you are testing and how you evaluated them (e.g. relevance of in-stream processes vs. other watershed compartments, such as downstream propagation or lateral inputs.

Ln 80. Just for curiosity, did you expect to observe differences between reaches or among sites? As it is written, it seems so; but you did not mention anything about that in the introduction nor discussion.

Ln 94. Longitudinal profiles were only used to validate the probe measurements, right? If so, I would simplify these sentences (i.e., "In addition, biweekly grab samples were collected at each site to validate probe measurements"). Also, it would be nice to show the uncertainty associated with these measurements.

Ln 97. How confident you are with your rating curve?

Ln 109. In my opinion, there is no need to use two travel times. I would use only nominal water residence time. However, the authors can easily convince me of the opposite.

Ln 111. Did you assume the same discharge at all sites? Is this assumption reasonable given the length of the stream section and the tributary? Also, why did you choose these widths?

Ln 115. I suggest to change the order of sections 2.3.1 and 2.3.2. First, you identified types of diel cycles; then, you investigated the processes involved in such patters. This suggestion also goes for the results section.

Ln 130. Did you analyze the relationship between the amplitude in diel variation of T,

S, h and stream NO3 concentration? May be worth to try.

Ln 134. Sorry, I did not follow this rationale. Several studies have related Cobs or Cres with diel patterns of environmental variables. Is it really necessary to use the first derivate? Using Cobs or Cres will simplify the results.

Figure 2. I would only plot those cases when r < 0.75 because, as you mentioned, cases with low r are difficult to interpret. If you do so, then you can color the data based on clusters. Finally, the caption should define all the elements (X-axis, legend, dashed horizontal line).

Ln 156. So, lag times (those with r > 0.75) are close to zero, but different from zero. Is that right? How do you explain it? Is it possible, then, that diel variations are a combination of in-stream processes and downstream propagation? Relatedly, have you check if the lag times vary across clusters? This may partially explain some of the observed patterns.

Ln 161. I missed some information in this section. For instance, which cluster dominates in each site? Some of this info is available in Figure 5, but should be more clearly stated here. Also, move Figure 5 here.

Figure 4. This figure has a lot of information and it is difficult to digest. Some ideas that came to my mind to improve it: (i) Panels A-C can be a table (Table 1). If you do so, then you can add some statistical test (e.g. Wilcoxon test) to show if clusters had different environmental conditions. (ii) Panels E-G can also be a table (Table 2). Here, you can report, for each cluster and relation, the mean r, the IQR of r, and the proportion of cases that has a significant relation (p-value < 0.05, or r > 0.5). In this way, the reader will easily see in which clusters these relations were consistent over time. (iii) It will be nice to show if there was a relationship between seasonal patterns of environmental variables and diel NO3 variability. If so, you can make a new figure showing these relations (similar to Fig 6 Heffernan and Cohen, or Fig. 6 Roberts and Mulholland 2007).

Figure 5. Given that the sensors were not allocated in all sites at the same time, perhaps it is better to report the relative values (e.g. days cluster 1/days with measurements) for each month. Also, I guess that the lack of values in S1 from April to June is due to missing data. Finally, it will be better to show the results in bars (not areas), as months is a discrete variable.

Ln 241. I agree that cluster F enclosed a wide range of diel NO3 patterns and environmental conditions; and thus, may be a box with all the "weird" days (i.e. storms). However, cluster E looks more consistent in terms of diel patterns and they may be related to in-stream processes (i.e. nitrification). My point here is that, based on your data, you cannot discard any hypothesis rather than longitudinal propagation; at least for clusters A-E.

Ln 243. Another possible explanation is that there is a lag time between light inputs and NO3 uptake (see Heffernan and Cohen 2010 discussion). A cross-correlation analysis can be a good way to test if there was a decoupling between light and stream NO3 concentration at daily scale.

Ln 252. Seasonal changes in light inputs occur even if there is no forest (i.e. the duration, timing and amount of sunlight varies over the year). Also, there are seasonal changes in the N demand by plants (see Heffernan and Cohen 2010).

Ln 258. Yes, phosphorous limitation may affect NO3 uptake. However, the relation N:P of this streams is < 16; suggesting that there is N limitation. Perhaps you don't need to go that far here (sometimes is better to keep the discussion simple and straightforward). One sentence stating that other factors, such as seasonal changes in nutrient availability, photoautotrophs stoichiometry, or temperature may further affect diel NO3 cycles is enough to make your point here.

Ln 278. Here, we are mixing apples with oranges. On one hand, some studies showed that diel patterns of NO3 concentrations changed during late-summer and fall, and that this phenomenon may be related to in-stream nitrification (e.g. Laursen 2004, Lupon

2016). The causes of this phenomenon is, as far as I know, under debate. It may be due to higher DOC inputs, or due to changes in pH and temperature. Curiously, this phenomenon seems to occur at S2 in September. On the other hand, Lupon 2020 showed that in-stream processes may vary along rivers. This may explain, for example, why S1 and S2 showed different diel patterns in September, or why the three sites did not show the same seasonal patterns. I would separate this two stories in two paragraph; one focused on in-stream processes and another one focused on why the three sites behave differently.

Technical notes

Ln 11. "sites" instead of "locations"

Ln 23. Better to say "in-stream processes can significantly influence loads and concentrations of nutrients". Further, Peterson et al. 2001 may be also a good, general reference for this sentence.

Ln 27: nitrogen (N)

Ln 27. Nitrate ($NO_3^-$). From hereafter, use $NO_3^-$ instead of nitrate.

Ln 32. "Carbon dioxide"

Ln 47 (and hereafter). The proper name of this process is "photoautotrophic uptake", not "autotrophic uptake" (nitrifiers are also autotrophs) nor "plant uptake" (mostly used for terrestrial systems). Also, the use of $U_a$ made sense in Cohen's papers, but not here. Use "photoautotrophic uptake" instead.

Ln 51. Nitpicking, but "microbial net depletion" sounds weird; perhaps "other in-stream processes"?

Ln 55. Same here. "Such diel variability in these other in-stream processes would cause..."

Ln 72. Technically, you are studying a stream section that is divided in two reaches.

Figure 1. The map should show the contributing catchment to S3. Also, I would delete the longitudinal profile, as you don't use this data in the current manuscript.

Ln 79-82. I would divide this sentence into two: one for each reach.

Ln. 80. Delete "and in this sense it (. . .) southwest Germany"

Ln 87. I missed some information about stream biotic compartments (e.g. emergent and floating macrophytes, algaes, biofilm). This is important to understand the role of photoautotrophic uptake.

Ln. 105. I would move this whole sentence to the introduction, when you state your expectations.

Ln 107. "patters, we determined (. . .) cross-correlation, which is (. . .)"

Ln 121. I understand why you named it "C residual". Yet, it may be more intuitive for the reader to refer it as "C corrected" or something like that.

Ln 129-141. Move this paragraph to the "Assessing the origin of diel nitrate variation" section.

Ln 137. This statement is not entirely true. Discharge can also affect in-stream processes (see Seybold and McGlynn 2016). Anyway, as I mentioned earlier, I would relate all environmental variables with Cres.

Ln 149. Nitpicking, but this heading does not seem right for the results. What about "Sources of diel patterns "?

Ln 152. Move this sentence to the methods section.

Ln. 168. Delete "a quarter of a period (0.5 travel time)"

Ln.169. Delete the whole sentence "Note that (. . .)."

Ln 171. Move everything related to drivers to another section and keep this one strictly to diel patterns characteristics.

Figure 3. Please, describe what the black dots and the shaded area represent (mean and standard deviation?).

Ln 214. Delete "However, (...) lag estimation."

Ln 223. What is the point of this paragraph? I might missed something. Do you mean that the observed diel pattern may be as a result of longitudinal propagation and in-stream processes?

Ln 241. "in-stream processes"

Ln 306. Clusters A and B, right?

––––––––––––––––––––––––––––––––––

---

## Referee Comment (RC2) · Anonymous Referee #2 · 4 Jan 2021

Review of "Diel patterns in nitrate concentration suggest importance of microbial"

Summary

Greiwe et al. collected diel nitrate data from three locations in a stream over multiple months to determine the controls of diel nitrate signals. They used cross correlation to show that diel signals were controlled by local in-stream processes rather than from upstream. Next, they used cluster analyses to identify consistent patterns in the diel signals. This is a novel and interesting approach. Finally, they relate the clusters with light and discharge to tease apart what is controlling each cluster.

I think this is an interesting and worthwhile paper. I particularly like the use of cluster analyses on the diel data to identify common trends in the diel cycle. However, I believe major revisions are necessary before publication. The biggest issue I have is the attempt to explain diel patterns based on unmeasured microbial processes. This is especially complicated given that many of these processes can cancel each other out (e.g., nitrification and denitrification) and we do not have easily measured proxies (like we have light for photosynthesis). Thus, I suggest that the authors tone down much of the speculation about microbial pathways, and instead focus on what they can show with data.

I also have some concerns with the methods. The cross-correlation approach could be described in more detail. Most importantly, there should be more detail about how the cluster analysis was performed. I am not an expert on cluster analyses and found it confusing how diel curves with multiple data points were put into a cluster analysis. As I mentioned above, I really liked this novel approach and I think it could be used for other constituents (DO, CO2, etc.). A better description of the methods would make it easier for others to replicate the analysis.

Title: I would remove the reference to microbial pathways. This paper has no data to back up the suggested trends in microbial processes.

Line 15: What is plug-flow?

Line 25: A key part of the spiral is that the nutrients are then mineralized to the water column to be taken up again downstream. This should be added here.

Line 31: Can you better describe the link between climate change and nutrient retention? What role does drought play?

Line 46: Denitrification is a heterotrophic process. This line implies that denit could occur via autotrophic processes. Please revise.

Line 93: Please provide more information about the periodical movement of the sensors. Were the sensors moved at equal intervals? Is the data available from each

sub-reach stratified across the sample period?

Lines 106-108: Could you provide more info on interpreting the cross-correlation data? What does a low and high correlation mean? How does this help better elucidate N transported from upstream vs. from in stream processes? A few lines here will help the reader going forward, especially to understand figure 2.

Line 110: What travel time distribution is this referring to? You only conducted one tracer release (I think).

Line 120: I believe that residual should be added earlier in the sentence. "...was done on the residuals of the diel solute concentration signal."

Lines 115-127: I am having troubles understanding how the clusters were determined, or in other words, how the k-means approach turned diel data into clusters. Could that be described more? I am used to clusters being used with single values (i.e., animal abundance data), so how can multiple points be used (i.e., from a diel curve). I do not have much experience with clustering, but that will be true for many readers as well. More detail would be helpful.

Line 160/Figure 2: This took some time to determine what I am looking at. Is the main point that points with a high cross-correlation are typically between 0 and the nominal travel time (the shaded area)? Either way I would add a line or two describing the main result out of this figure. Also, how is it possible that a travel time is negative?

Line 175/Figure 3: Do the shaded areas represent a confidence interval? And what calculations were used to calculate the shaded area?

Line 200/Figure 5: Would it be logical to make the y-axis a proportion? The ups and downs are distracting. Making them a proportion would better show the seasonal trends.

Line 201: Something is missing here. Maybe, "Relation of nitrate clusters and reach balance"

Line 220: Please define or further explain short-circuiting.

Line 223: "stated"? Maybe observed?

Line 226: This explanation of the Hensley and Cohen paper is confusing and hard to follow. Could you describe the point of the paper without getting into the details?

Line 240: I don't believe the description of clusters E and F being influenced by discharge is in the results section. How did you come to this conclusion?

Line 260: What is the relevance of the 0.5 mg/L SRP? What does this threshold indicate?

Line 287: This is also true for estimates of stream metabolism.

Line 247-300: There is a lot of speculation on the drivers of diel patterns in here. It would be much more convincing to use a statistical analyses/models to make conclusions about what is controlling the diel trends rather than relying on the literature and instinct. The correlations with light are somewhat compelling for the first two clusters but it is still hard to disentangle the different microbial pathways relative to the autotrophic. For the other clusters it gets much more complicated and interpretation is pure speculation. That being said, I still think these data are useful and novel. But tying each cluster to a specific driver is for another paper in my opinion. I suggest that this part of the discussion be substantially shortened. I like how you first describe the strong evidence that in-stream, not upstream, processes are driving diel trends. Then go through the clusters or sets of clusters and do some light speculation on the drivers of the signals in relation to the literature. This is done quite well in lines 303-331.

Line 353: Is there a citation for these data?

Line 355: The topic of groundwater should be introduced and described much earlier in the methods section. Also, please address how groundwater might affect the diel curves? Groundwater is likely an important factor for diel curves during summer low flows.

Line 371: We know this already–In my opinion, this is not the strength of this paper. I would end here with a line noting how you were able to separate diel trends in NO3 con-centrations into clear clusters with distinct diel patterns and probably different drivers. These clusters can be used a blueprint for future efforts to model drivers of N cycling. Likewise, using the cluster analyses on diel data is a novel approach and could be used for other measurements (e.g., DO, CO2, SRP, etc).

---

## Referee Comment (RC3) · Anonymous Referee #3 · 5 Jan 2021

This novel approach of analyzing and visualizing diel nutrient data is an important contribution to stream ecosystem science. It fits the scope of this journal well. Overall, I found this manuscript to be interesting and advancing the use of diel cycles of nutrients to interpret ecological functions in streams. However, the lack of simultaneously measured process rates (such as metabolism, nitrification or denitrification) makes parts of the discussion and conclusions very speculative and I strongly recommend to shorten and nuance that section.

Specific comments: 1. In figure 1 I wonder why the evaluated stream reach is mapped outside of the land use map? In particular, information on urban areas including pasture

between the measuring points are of interest to the interpretation of this data set.

2. Line 220. "Downstream transport of solute signals therefore fails to explain most of our data. We therefore interpret our data to indicate primarily in-stream origin of diel nitrate cycles." What about signals from land, i.e. soil water signals. Especially during low flow. I realize this comes later in the manuscript but I would move some of that discussion here and clarify it also in the methods.

3. Line 368-60 "In the remaining clusters temporal shifts were evident that could be explained by temporal shifts in microbial nitrate processing but not by photosynthesis-driven uptake." This line makes it sound like you measured microbial processing or photosynthesis, please re-phrase.

4. Line 250-256. My experience of dissolved oxygen signals is that they can often match cluster C, with maximum %O2 in the afternoon. I would not be so quick to discard cluster C from being driven by photoautotrophs without evidence. Especially since there was a negative correlation between solar radiation and cluster C (line 183), which is what you use to argue for photoautotrophic dominance in driving cluster A and B.

5. Could spring photoautotrophs be light inhibited during mid-day and therefore cluster C peaks in the afternoon? Cluster C was most prominent in spring when harmful UV is the highest. Which were the light levels in this study? Was light ever measured under water?

6. No statistics are presented in the results section on page 8, please include that.

---

## Author Comment (AC1) · 25 Jan 2021

**Author's response to comments by anonymous referee #1**

We would like to thank referee #1 for taking the time to read our manuscript carefully and for providing very constructive feedback. The helpful suggestions will improve the quality of our manuscript. Our responses to the individual comments are shown in blue below.

**Summary**

This paper examines patterns and sources of diel variation in stream NO3 concentration along a lowland river in Germany. The authors show that diel patterns of stream NO3 concentration vary over the growing season, yet most days show similar diurnal oscillations. Further, by combining different statistical techniques, the authors convincingly show that diel patterns are mostly driven by in-stream processes. Finally, the authors analyze diel and seasonal patterns of several environmental variables to discuss which in-stream process is driving diel NO3 cycles.

**General comments**

This paper makes a significant contribution to watershed and stream ecology through its assessment of patterns and controls of diel variation in stream NO3 concentration. However, I have some major issues that need to be addressed. All comments are made in the spirit of increasing the potential impact of this interesting research.

1. While most of the findings presented in the paper are original and compelling, the conclusions raised from them are sometimes speculative and inaccurate. For instance, the authors concluded that "the magnitude of microbial NO3 processing may be large compared to plant uptake", but they did not measure any in-stream process (GPP, denitrification, nitrification) nor NO3 uptake rates. Hence, it is impossible to know, based on their data and results, which in-stream process was contributing the most to NO3 uptake rates over the study period. Similarly, they stated that "diel patterns in NO3 concentration suggest the importance of microbial pathways for in-stream processing", but the 70% of diel patterns seem to be driven by photoautotrophic uptake (not microbial pathways). My suggestion is to focus the objectives and conclusions on the compelling results and only speculate about the relative importance of different in-stream processes in the discussion.

> Reply: We agree that our statements on the relative importance of biochemical processes require some assumptions and should not be presented as key findings. We are happy to shorten that section and shift the focus of this paper towards the role of in-stream and transport processes as suggested by referee #1 below.

2. I missed some results regarding lateral inputs. In the discussion, the authors mentioned that lateral inputs may not affect diel NO3 patterns because they did not observe diel variations in discharge. While I agree with this statement, lateral inputs should be included in the hypothesis, methods and results (see Flewelling et al. 2014 or Lupon et al. 2016). Also, the authors mentioned that there was a tributary entering to the upstream reach. Does the tributary show diel variation in NO3 concentration? How this may influence stream NO3 concentration in S2?

> Reply: Unfortunately, we lack information about diel patterns in the tributary. In a revised manuscript, we will discuss the potential impact of lateral inputs and diel variation in the tributary based on its size and the transformation of the concentration signal between S1 and S2 (the sampling points between which the tributary enters the main stream).

3. I was confused by some of the approaches used. For instance, what is the point of the mass balance? It has many uncertainties (e.g. groundwater, tributaries) and the results derived from it are difficult to interpret. My suggestion is to delete this whole section. Instead, I will focus on analyzing (i) if all sites showed similar seasonal patterns in diel NO3 variation (i) if the effect of longitudinal propagation differed across clusters; (iii) if there was a lag time between diel patterns of drivers and stream NO3 concentration (see my specific comments for more info on this regard).

Reply: We agree that the mass balance has many uncertainties. Considering our shifted focus, the mass balance will be less important for our main statements. Nevertheless, mass balances are important tools in hydrology and readers may expect to find it in our manuscript given the experimental setup. In order to provide a complete analysis we will include it as supplementary material. We are grateful for the suggestions regarding data analysis. In the original manuscript we already compare seasonal patterns (i) at the monitoring sites in figure 5. The effect of longitudinal propagation (ii) will be assessed for the different clusters in a revised manuscript by modifying figure 2 as suggested by referee #1 below. A lag between drivers and concentration (iii) was in fact observed and is evident in the clusters we found. While timing of irradiance and stream temperature was more or less constant throughout the year, the nitrate clusters reflect different lags relative to these drivers. We will revise the manuscript to make this more clearly.

4. The discussion is a little bit puzzling. My suggestion is to delete all sub-headings and focus on how different sources shape stream NO3 concentration. You can start with a paragraph discarding longitudinal propagation and lateral inputs as factors causing diel NO3 patterns. Then, move to the most obvious process: photoautotrophic uptake (clusters A-B) and how it varies over time depending on light, temperature, discharge. Finally, you can suggest potential explanations for the other clusters: denitrification (cluster C), nitrification (cluster D), storm flow (cluster F).

Reply: We appreciate this suggestion and will revise our discussion accordingly.

5. While I like the figures, most of them (and their captions) need some improvements (see my specific comments). Also, I missed a figure showing the raw data (i.e. diel patterns of NO3, discharge, light and temperature over the whole study period). This figure is key to understand some of the points discussed (e.g. no diel variation in discharge); and it will be very helpful to the readers.

Reply: The figures will be revised according to the comments below and a figure showing the raw data will be added.

**Specific comments**

Ln 1. The title is a little bit speculative. Perhaps something focused on in-stream processes vs longitudinal propagation would be better.

Reply: We will change the title to "Diel patterns in nitrate concentration produced by in-stream processes".

Ln 21. This sentence is not accurate. What your results are telling us is that different in-stream processes might generate diel patterns in NO3 concentration, and that the relative importance of such processes may vary depending on the season.

Reply: This sentence will be deleted.

Ln 37-44. This rationale is correct, but does not engage with the objective of the paper (i.e. you don't quantify any in-stream process). My suggestion is to delete this part and merge this paragraph with the following one.

Reply: We agree that this section should be shortened and merged with the following.

Ln 46-55. As it occurs with the previous paragraph, this section goes beyond the objectives of the paper. My suggestion here is to shorten it to something like "Previous studies have suggested that seasonal patterns of diel variation in stream NO3 concentration are related to in-stream photoautotrophic uptake (refs). Due to photosynthetic light requirements, photoautotrophs take up NO3 mostly during the day, with minimum and maximum NO3 concentrations occurring at X and Y (refs). However, there is evidence that diel variation (: : :)."

Reply: We agree that this and the previous paragraph include unnecessary information and will merge them in order to streamline the introduction.

Ln 67. What is the difference between the two hypothesis? They look exactly the same to me. Be explicit with the hypotheses you are testing and how you evaluated them (e.g. relevance of in-stream processes vs. other watershed compartments, such as downstream propagation or lateral inputs.

> Reply: We thank referee #1 for pointing out that our hypotheses need improvement. Rethinking this, we decided to take up the suggestion by referee #1 below. This means we move the pattern identification to the front and then test possible explanations. Our research questions would then be: What patterns in nitrate concentration do we find in the studied system? Are these patterns produced by in-stream processes or transport? And, finally, how are they related to drivers of biochemical processes?

Ln 80. Just for curiosity, did you expect to observe differences between reaches or among sites? As it is written, it seems so; but you did not mention anything about that in the introduction nor discussion.

> Reply: In fact, we expected to see differences between the two reaches as a result of the revitalization measures in the downstream reach. However, during the course of the monitoring it became more and more obvious that it would be extremely challenging to separate these effects from other influences (e.g. groundwater, tributary). We think we should include these expectations in the introduction and state that we were unable to separate potential effects of the revitalization from other influences.

Ln 94. Longitudinal profiles were only used to validate the probe measurements, right? If so, I would simplify these sentences (i.e., "In addition, biweekly grab samples were collected at each site to validate probe measurements"). Also, it would be nice to show the uncertainty associated with these measurements.

> Reply: The reference to longitudinal concentration profiles is, indeed, unnecessary and will be removed.

Ln 97. How confident you are with your rating curve?

> Reply: The rating curve could have more data points – as always - but is considered sufficient to provide the discharge corresponding to our water level measurements. The validity of the rating curve is limited to the range of the tracer injections and to the corresponding monitoring site (S3). Despite its uncertainty, our field rating curve certainly reflects reality much better than discharge data from a official gauging station tens of km upstream.

Ln 109. In my opinion, there is no need to use two travel times. I would use only nominal water residence time. However, the authors can easily convince me of the opposite.

> Reply: Maybe using nominal residence time would be sufficient. However, nominal residence time strongly depends on channel geometry and discharge which both are subject to uncertainties. In fact, both independent approaches produce similar results, which increases our confidence in travel time estimates and we would like to keep both in the revised manuscript.

Ln 111. Did you assume the same discharge at all sites? Is this assumption reasonable given the length of the stream section and the tributary? Also, why did you choose these widths?

> Reply: In fact, our discharge measurements at S3 may not be representative for the entire stream section due to unknown contributions from the tributary (which we estimate based on visual judgement to account for a maximum 10 % of stream flow) and possibly groundwater. These uncertainties mainly impact our mass balances. In the remaining part of our analysis discharge or water level data from S3 is used to generally characterize discharge conditions.
>
> Channel widths are roughly based on aerial imagery. However, channel geometry has further sources of uncertainty (e.g. distribution of water depth and discharge) that we were unable to accurately account for. For estimating nominal travel time, we therefore selected a generous range of possible stream widths and used independent travel time estimates from our tracer injection to validate our assumptions.

Ln 115. I suggest to change the order of sections 2.3.1 and 2.3.2. First, you identified types of diel cycles; then, you investigated the processes involved in such patters. This suggestion also goes for the results section.

Reply: We appreciate this suggestion and will restructure the manuscript accordingly.

Ln 130. Did you analyze the relationship between the amplitude in diel variation of T, S, h and stream NO3 concentration? May be worth to try.

Reply: Good point. In order to assign clusters to drivers, it would be nice to show that these variables are not only related to nitrate concentration in terms of timing (which we have shown already) but also in terms of magnitude. We actually did that, but amplitudes of environmental variables were not clearly correlated with nitrate concentrations. We will mention this in the revised manuscript.

Ln 134. Sorry, I did not follow this rationale. Several studies have related Cobs or Cres with diel patterns of environmental variables. Is it really necessary to use the first derivate? Using Cobs or Cres will simplify the results.

Reply: We are indeed convinced that drivers of biochemical processes are only indirectly related to solute concentrations. If we e.g. assume irradiance as the main driver of photosynthesis and associated nitrate assimilation, we would expect the strongest effect on nitrate concentrations, when irradiance is strongest (ignoring that increasing irradiance may actually inhibit photosynthesis at some point). Or in other words, maximum irradiance would coincide with maximum rate of change (i.e. its first derivative) of the nitrate concentration. The idea that drivers influence the change of concentrations is also common in solute models, e.g. consider Hensley and Cohen (2016) for the case of nitrate or Grace et al. (2015) for the case of dissolved oxygen.

Figure 2. I would only plot those cases when r < 0.75 because, as you mentioned, cases with low r are difficult to interpret. If you do so, then you can color the data based on clusters. Finally, the caption should define all the elements (X-axis, legend, dashed horizontal line).

Reply: Good idea. We will revise Figure 2 accordingly.

Ln 156. So, lag times (those with r > 0.75) are close to zero, but different from zero. Is that right? How do you explain it? Is it possible, then, that diel variations are a combination of in-stream processes and downstream propagation? Relatedly, have you check if the lag times vary across clusters? This may partially explain some of the observed patterns.

Reply: We will include a comparison of time lags across clusters – which we did already but which was not included in the original manuscript. The interpretation of this finding is a bit challenging. It is clear that both processes (biochemical processing and transport) occur simultaneously as water travels downstream but it is not clear how exactly this produces the lags. If the signal was simply advected, lags should be at least as long as nominal travel time, probably longer (due to dispersion). Values clearly below travel time cannot be explained by advection. Lags may be influences by sudden changes in discharge conditions, e.g. a sudden flood wave may cause the downstream advection of a longitudinal pattern previously created by biochemical processes during a low flow period. However, our data do not show a clear water level threshold that separates in-stream control from transport control of diel nitrate patterns. We will add this aspect to the discussion.

Ln 161. I missed some information in this section. For instance, which cluster dominates in each site? Some of this info is available in Figure 5, but should be more clearly stated here. Also, move Figure 5 here.

Reply: We thank referee #1 for the suggestion to also assess longitudinal stability/transformation of the clusters and will include this aspect in a revised manuscript.

Figure 4. This figure has a lot of information and it is difficult to digest. Some ideas that came to my mind to improve it: (i) Panels A-C can be a table (Table 1). If you do so, then you can add some

statistical test (e.g. Wilcoxon test) to show if clusters had different environmental conditions. (ii) Panels E-G can also be a table (Table 2). Here, you can report, for each cluster and relation, the mean r, the IQR of r, and the proportion of cases that has a significant relation (p-value < 0.05, or r > 0.5). In this way, the reader will easily see in which clusters these relations were consistent over time. (iii) It will be nice to show if there was a relationship between seasonal patterns of environmental variables and diel NO3 variability. If so, you can make a new figure showing these relations (similar to Fig 6 Heffernan and Cohen, or Fig. 6 Roberts and Mulholland 2007).

> Reply: We agree that figure 4 contains much information. However, we are convinced that this information is more easily accessible to the reader when presented as figures rather than as tables. In order to make this information more digestible we suggest to split figure 4 into two figures, one showing daily values of environmental parameters and the other one showing daily correlations.

Figure 5. Given that the sensors were not allocated in all sites at the same time, perhaps it is better to report the relative values (e.g. days cluster 1/days with measurements) for each month. Also, I guess that the lack of values in S1 from April to June is due to missing data. Finally, it will be better to show the results in bars (not areas), as months is a discrete variable.

> Reply: We agree with the bars but not with the relative values. Relative values may suggest information were we actually have missing data. It is transparent for the reader if we keep showing absolute values. We will add an explanation to the caption that columns with less than 30 or 31 days are due to missing data.

Ln 241. I agree that cluster F enclosed a wide range of diel NO3 patterns and environmental conditions; and thus, may be a box with all the "weird" days (i.e. storms). However, cluster E looks more consistent in terms of diel patterns and they may be related to in-stream processes (i.e. nitrification). My point here is that, based on your data, you cannot discard any hypothesis rather than longitudinal propagation; at least for clusters A-E.

> Reply: We agree that discharge during cluster E was not sufficiently different from clusters A,C and D to justify discarding in-stream processes. We will revise this section and also consider in-stream processes.

Ln 243. Another possible explanation is that there is a lag time between light inputs and NO3 uptake (see Heffernan and Cohen 2010 discussion). A cross-correlation analysis can be a good way to test if there was a decoupling between light and stream NO3 concentration at daily scale.

> Reply: To be precise, Heffernan and Cohen discuss the lag between photosynthesis (not instantaneous light availability) and nitrate assimilation. However, as mentioned above, a lag between light and assimilation is expected because light as a driver of assimilation influences the rate of change (i.e. the derivative) of both nitrate and dissolved oxygen concentration. We thank referee #1 for the suggestion to use cross-correlation to analyse this effect. The lag between variable diel patterns in nitrate concentration and comparatively stable diel patterns in drivers is also reflected in the results of the cluster analysis. We, therefore, do not see a real advantage in using a different methodology to show the same effect. However, we will refer to the lag between drivers and concentrations more explicitly in a revised manuscript.

Ln 252. Seasonal changes in light inputs occur even if there is no forest (i.e. the duration, timing and amount of sunlight varies over the year). Also, there are seasonal changes in the N demand by plants (see Heffernan and Cohen 2010).

> Reply: True, this information should will be added to the discussion.

Ln 258. Yes, phosphorous limitation may affect NO3 uptake. However, the relation N:P of this streams is < 16; suggesting that there is N limitation. Perhaps you don't need to go that far here (sometimes is better to keep the discussion simple and straightforward). One sentence stating that other factors, such as seasonal changes in nutrient availability, photoautotrophs stoichiometry, or temperature may further affect diel NO3 cycles is enough to make your point here.

> Reply: We agree and will revise this section accordingly.

Ln 278. Here, we are mixing apples with oranges. On one hand, some studies showed that diel patterns of NO3 concentrations changed during late-summer and fall, and that this phenomenon may be related to in-stream nitrification (e.g. Laursen 2004, Lupon 2016). The causes of this phenomenon is, as far as I know, under debate. It may be due to higher DOC inputs, or due to changes in pH and temperature. Curiously, this phenomenon seems to occur at S2 in September. On the other hand, Lupon 2020 showed that in-stream processes may vary along rivers. This may explain, for example, why S1 and S2 showed different diel patterns in September, or why the three
sites did not show the same seasonal patterns. I would separate this two stories in two paragraph; one focused on in-stream processes and another one focused on why the three sites behave differently.

> Reply: We agree and will revise this section to make it less confusing.

**Technical notes**

Ln 11. "sites" instead of "locations"

> Reply: We will replace "locations" by "sites".

Ln 23. Better to say "in-stream processes can significantly influence loads and concentrations of nutrients". Further, Peterson et al. 2001 may be also a good, general reference for this sentence.

> Reply: We will revise this sentence and add the recommended reference.

Ln 27: nitrogen (N)

> Reply: We will revise the use of abbreviations throughout the manuscript.

Ln 27. Nitrate (NO3-). From hereafter, use NO3- instead of nitrate.

> Reply: s. above

Ln 32. "Carbon dioxide"

> Reply: s. above

Ln 47 (and hereafter). The proper name of this process is "photoautotrophic uptake", not "autotrophic uptake" (nitrifiers are also autotrophs) nor "plant uptake" (mostly used for terrestrial systems). Also, the use of Ua made sense in Cohen's papers, but not here. Use "photoautotrophic uptake" instead.

> Reply: We will check and revise terminology throughout the manuscript.

Ln 51. Nitpicking, but "microbial net depletion" sounds weird; perhaps "other in-stream processes"?

> Reply: We will replace "microbial net depletion" by "other in-stream processes".

Ln 55. Same here. "Such diel variability in these other in-stream processes would cause: : :"

> Reply: s. above

Ln 72. Technically, you are studying a stream section that is divided in two reaches.

> Reply: We will use the terms "stream section" and "upper/lower reach" in the revised manuscript.

Figure 1. The map should show the contributing catchment to S3. Also, I would delete the longitudinal profile, as you don't use this data in the current manuscript.

> Reply: The map will be revised accordingly.

Ln 79-82. I would divide this sentence into two: one for each reach.

> Reply: The sentence will be divided.

Ln. 80. Delete "and in this sense it (: : :) southwest Germany"

> Reply: This will be corrected.

Ln 87. I missed some information about stream biotic compartments (e.g. emergent and floating macrophytes, algaes, biofilm). This is important to understand the role of photoautotrophic uptake.

Reply: Information about biotic compartments will be added.

Ln. 105. I would move this whole sentence to the introduction, when you state your expectations.
Reply: This will be corrected.

Ln 107. "patters, we determined (: : :) cross-correlation, which is (: : :)"
Reply: This sentence will be simplified.

Ln 121. I understand why you named it "C residual". Yet, it may be more intuitive for the reader to refer it as "C corrected" or something like that.
Reply: Not sure, if 'corrected' better describes what we did. We suggest to call it "C diel".

Ln 129-141. Move this paragraph to the "Assessing the origin of diel nitrate variation" section.
Reply: This will be done as part of our restructuring of the manuscript.

Ln 137. This statement is not entirely true. Discharge can also affect in-stream processes (see Seybold and McGlynn 2016). Anyway, as I mentioned earlier, I would relate all environmental variables with Cres.
Reply: We agree that the phrasing may be misunderstood. We did not mean to exclude the impact of water level and discharge on biochemical process and will revise this section accordingly.

Ln 149. Nitpicking, but this heading does not seem right for the results. What about "Sources of diel patterns "?
Reply: The heading will be revised accordingly.

Ln 152. Move this sentence to the methods section.
Reply: The sentence will be moved to the methods section.

Ln. 168. Delete "a quarter of a period (0.5 travel time)"
Reply: This will be deleted.

Ln.169. Delete the whole sentence "Note that (: : :)."
Reply: The sentence will be deleted.

Ln 171. Move everything related to drivers to another section and keep this one strictly to diel patterns characteristics.
Reply: We are happy to do so, particularly as figure 4, to which these lines refer, will be split into two (s. above).

Figure 3. Please, describe what the black dots and the shaded area represent (mean and standard deviation?).
Reply: This information will be added to the figure.

Ln 214. Delete "However, (: : :) lag estimation."
Reply: This sentence will become obsolete anyway as the information necessary for the interpretation of figure 2 will be provided in the methods section.

Ln 223. What is the point of this paragraph? I might missed something. Do you mean that the observed diel pattern may be as a result of longitudinal propagation and in-stream processes?
Reply: Here we attempt to interpret our findings regarding in-stream vs. transport control of diel nitrate cycles. The paragraph will be revised to make it more understandable.

Ln 241. "in-stream processes"
Reply: This will be corrected.

Ln 306. Clusters A and B, right?

Reply: Yes, this will be revised accordingly.

**References**

Grace, M. R., Giling, D. P., Hladyz, S., Caron, V., Thompson, R. M., and Mac Nally, R.: Fast processing of diel oxygen curves: Estimating stream metabolism with BASE (BAyesian Single-station Estimation), Limnol. Oceanogr. Methods, 13, e10011, doi:10.1002/lom3.10011, 2015.

Hensley, R. T. and Cohen, M. J.: On the emergence of diel solute signals in flowing waters, Water Resour. Res., 52, 759–772, doi:10.1002/2015WR017895, 2016.

---

## Author Comment (AC2) · 25 Jan 2021

**Author's response to comments by anonymous referee #2**

We would like to thank referee #2 for taking the time to read our manuscript carefully and for providing constructive feedback. The helpful suggestions will improve the quality of our manuscript. Our responses to the individual comments are shown in blue below.

**Summary**

Greiwe et al. collected diel nitrate data from three locations in a stream over multiple months to determine the controls of diel nitrate signals. They used cross correlation to show that diel signals were controlled by local in-stream processes rather than from upstream. Next, they used cluster analyses to identify consistent patterns in the diel signals. This is a novel and interesting approach. Finally, they relate the clusters with light and discharge to tease apart what is controlling each cluster. I think this is an interesting and worthwhile paper. I particularly like the use of cluster analyses on the diel data to identify common trends in the diel cycle. However, I believe major revisions are necessary before publication. The biggest issue I have is the attempt to explain diel patterns based on unmeasured microbial processes. This is especially complicated given that many of these processes can cancel each other out (e.g., nitrification and denitrification) and we do not have easily measured proxies (like we have light for photosynthesis). Thus, I suggest that the authors tone down much of the speculation about microbial pathways, and instead focus on what they can show with data.

> Reply: We appreciate the positive attitude of referee #2. We agree that our statements on the relative importance of processes are too speculative and should not be presented as key findings and will significantly shorten this section in a revised version of this manuscript. This has also been suggested by referee #1.

I also have some concerns with the methods. The cross-correlation approach could be described in more detail. Most importantly, there should be more detail about how the cluster analysis was performed. I am not an expert on cluster analyses and found it confusing how diel curves with multiple data points were put into a cluster analysis. As I mentioned above, I really liked this novel approach and I think it could be used for other constituents (DO, CO2, etc.). A better description of the methods would make it easier for others to replicate the analysis.

> Reply: We will add information on the methodology to make our analysis more understandable. The description of the cluster method was intentionally kept short as it was applied before by Aubert and Breuer (2016) (L. 117).

Title: I would remove the reference to microbial pathways. This paper has no data to back up the suggested trends in microbial processes.

> Reply: The title will be changed to "Diel patterns in nitrate concentration produced by in-stream processes".

Line 15: What is plug-flow?

> Reply: The concept of plug flow originates from fluid mechanics where it describes uniform velocity profiles in pipes. In this concept zero dispersion and zero storage zone exchange are assumed. We will introduce the concept in a revised manuscript.

Line 25: A key part of the spiral is that the nutrients are then mineralized to the water column to be taken up again downstream. This should be added here.

> Reply: We will add this aspect to the description of 'nutrient spiraling'.

Line 31: Can you better describe the link between climate change and nutrient retention? What role does drought play?

> Reply: Climate change increases the probability of drought (at least these are the predictions for Southwest Germany) causing low flow conditions or zero flow (as in the reference) to occur more frequently. This in turn may influence stream temperature, light penetration and

other aspects that are important for biochemical in-stream processes. We will explain this in more detail in a revised manuscript.

Line 46: Denitrification is a heterotrophic process. This line implies that denit could occur via autotrophic processes. Please revise.

Reply: We will revise this phrase to prevent misunderstanding.

Line 93: Please provide more information about the periodical movement of the sensors. Were the sensors moved at equal intervals? Is the data available from each sub-reach stratified across the sample period?

Reply: In this figure it will be visible which sensor was placed where for how long.

Lines 106-108: Could you provide more info on interpreting the cross-correlation data? What does a low and high correlation mean? How does this help better elucidate N transported from upstream vs. from in stream processes? A few lines here will help the reader going forward, especially to understand figure 2.

Reply: We thank referee #2 for pointing out that we provided insufficient information on our methods. We will provide more background information in a revised manuscript.

Line 110: What travel time distribution is this referring to? You only conducted one tracer release (I think).

Reply: Exactly, that is the one we are referring to here. So this only gave us one data point (black cross in Fig. 2). In order to visualize how travel time would change when discharge changes we additionally estimated a range of nominal travel time estimates based on the local channel geometry.

Line 120: I believe that residual should be added earlier in the sentence. ": : :was done on the residuals of the diel solute concentration signal."

Reply: Thank you. This will in fact improve readability and will be changed accordingly.

Lines 115-127: I am having troubles understanding how the clusters were determined, or in other words, how the k-means approach turned diel data into clusters. Could that be described more? I am used to clusters being used with single values (i.e., animal abundance data), so how can multiple points be used (i.e., from a diel curve). I do not have much experience with clustering, but that will be true for many readers as well. More detail would be helpful.

Reply: We will provide additional information on that to make the methods section more understandable.

Line 160/Figure 2: This took some time to determine what I am looking at. Is the main point that points with a high cross-correlation are typically between 0 and the nominal travel time (the shaded area)? Either way I would add a line or two describing the main result out of this figure. Also, how is it possible that a travel time is negative?

Reply: Thank you. We realize that this figure requires more explanation which will be added in a revised manuscript. Of course, travel time cannot be negative, but lags determined by cross-correlation can. In our study this occurred when cross-correlation between the two signals was weak, i.e. the signals were not very similar. We agree that these data points are confusing and distract from the key massage of this figure. As suggested by referee #1 we will therefore remove all data points representing non-similar signals ($r > 0.75$).

Line 175/Figure 3: Do the shaded areas represent a confidence interval? And what calculations were used to calculate the shaded area?

Reply: Shaded areas in figure 3 represent the range between the 5th and 95th percentile, i.e. 90% of the data and black dots represent medians. We will add this information to the figure.

Line 200/Figure 5: Would it be logical to make the y-axis a proportion? The ups

and downs are distracting. Making them a proportion would better show the seasonal trends.

> Reply: We agree that the ups and downs are somewhat distracting. On the other hand, showing relative data may suggest that there is information where we actually have none. We therefore opt to keep the absolute values but represent months as bars (not areas) as suggested by referee #1.

Line 201: Something is missing here. Maybe, "Relation of nitrate clusters and reach balance"

> Reply: Thank you. This will be corrected.

Line 220: Please define or further explain short-circuiting.

> Reply: This phrasing will be revised.

Line 223: "stated"? Maybe observed?

> Reply: This phrasing will be revised.

Line 226: This explanation of the Hensley and Cohen paper is confusing and hard to follow. Could you describe the point of the paper without getting into the details?

> Reply: As referee #1 had similar concerns regarding this paragraph, we will do our best to simplify this section and make it more understandable.

Line 240: I don't believe the description of clusters E and F being influenced by discharge is in the results section. How did you come to this conclusion?

> Reply: We indeed missed to state that in the results section. The fact that discharge during cluster F (n=5) was elevated can be seen in figure 4. We may have been a bit quick stating the same for cluster E (n=21). Anyway, we should be careful not to overinterpret clusters with only few data points.

Line 260: What is the relevance of the 0.5 mg/L SRP? What does this threshold indicate?

> Reply: Around this threshold we would be able to detect SRP via ion chromatography. However, we are aware that such values would be quite high for surface waters in the study region. This threshold is not of any biological relevance and we will delete this sentence from the manuscript.

Line 287: This is also true for estimates of stream metabolism.

> Reply: In fact, stream metabolism estimated from dissolved oxygen data relies on very similar assumptions.

Line 247-300: There is a lot of speculation on the drivers of diel patterns in here. It would be much more convincing to use a statistical analyses/models to make conclusions about what is controlling the diel trends rather than relying on the literature and instinct. The correlations with light are somewhat compelling for the first two clusters but it is still hard to disentangle the different microbial pathways relative to the autotrophic. For the other clusters it gets much more complicated and interpretation is pure speculation. That being said, I still think these data are useful and novel. But tying each cluster to a specific driver is for another paper in my opinion. I suggest that this part of the discussion be substantially shortened. I like how you first describe the strong evidence that in-stream, not upstream, processes are driving diel trends. Then go through the clusters or sets of clusters and do some light speculation on the drivers of the signals in relation to the literature. This is done quite well in lines 303-331.

> Reply: We agree that parts of our discussion are very speculative and are not sufficiently backed up with data and our analysis. We will shorten corresponding parts of the discussion.

Line 353: Is there a citation for these data?

> Reply: The corresponding groundwater data are publicly available. A and figure and references will be provided in the supplementary material.

Line 355: The topic of groundwater should be introduced and described much earlier in the methods section. Also, please address how groundwater might affect the diel curves? Groundwater is likely an important factor for diel curves during summer low flows.

> Reply: We will introduce the potential influence of groundwater earlier in the manuscript.

Line 371: We know this already–In my opinion, this is not the strength of this paper. I would end here with a line noting how you were able to separate diel trends in NO3 concentrations into clear clusters with distinct diel patterns and probably different drivers.
These clusters can be used a blueprint for future efforts to model drivers of N cycling.
Likewise, using the cluster analyses on diel data is a novel approach and could be used for other measurements (e.g., DO, CO2, SRP, etc).

> Reply: We appreciate these suggestions and we will revise the conclusion in accordance with the new focus of the paper.

---

## Author Comment (AC3) · 25 Jan 2021

**Author's response to comments by anonymous referee #3**

We would like to thank referee #3 for taking the time to read our manuscript and for providing constructive feedback. The helpful comments will improve the quality of our manuscript. Our responses to the individual comments are shown in blue below.

This novel approach of analyzing and visualizing diel nutrient data is an important contribution to stream ecosystem science. It fits the scope of this journal well. Overall, I found this manuscript to be interesting and advancing the use of diel cycles of nutrients to interpret ecological functions in streams. However, the lack of simultaneously measured process rates (such as metabolism, nitrification or denitrification) makes parts of the discussion and conclusions very speculative and I strongly recommend to shorten and nuance that section.

> Reply: We agree that our statements on the relative importance are somewhat speculative and should not be presented as key findings. We will shorten the corresponding parts of the discussion and focus on compelling results.

Specific comments:

1. In figure 1 I wonder why the evaluated stream reach is mapped outside of the land use map? In particular, information on urban areas including pasture between the measuring points are of interest to the interpretation of this data set.

> Reply: Figure 1 will be revised accordingly.

2. Line 220. "Downstream transport of solute signals therefore fails to explain most of our data. We therefore interpret our data to indicate primarily in-stream origin of diel nitrate cycles." What about signals from land, i.e. soil water signals. Especially during low flow. I realize this comes later in the manuscript but I would move some of that discussion here and clarify it also in the methods.

> Reply: The topic of groundwater/soil water will be moved towards the beginning of the manuscript.

3. Line 368-60 "In the remaining clusters temporal shifts were evident that could be explained by temporal shifts in microbial nitrate processing but not by photosynthesisdriven uptake." This line makes it sound like you measured microbial processing or photosynthesis, please re-phrase.

> Reply: We will rephrase this sentence to avoid misunderstanding.

4. Line 250-256. My experience of dissolved oxygen signals is that they can often match cluster C, with maximum %O2 in the afternoon. I would not be so quick to discard cluster C from being driven by photoautotrophs without evidence. Especially since there was a negative correlation between solar radiation and cluster C (line 183), which is what you use to argue for photoautotrophic dominance in driving cluster A and B.

> Reply: In fact, there is no sharp border between clusters. Cluster C may contain days where drivers deviated from the usual pattern. However, this was not systematically the case as correlations of cluster C with radiation were weaker than those of cluster B and (slightly) cluster A. We will refer to this in the discussion.

5. Could spring photoautotrophs be light inhibited during mid-day and therefore cluster C peaks in the afternoon? Cluster C was most prominent in spring when harmful UV is the highest. Which were the light levels in this study? Was light ever measured under water?

> Reply: We will include this question in the revised discussion. Light was only measured as global irradiance at a nearby climate station and never under water.

6. No statistics are presented in the results section on page 8, please include that.

Reply: Statistics will be added to the results on page 8.

---

## Referee Comment (RC4) · Anonymous Referee #4 · 3 Feb 2021

Summary The manuscript by Greiwe et al. describes a spatially-repeated sampling of diel variation in nitrate export along a reach in an intermediate watershed. The authors collected high-frequency diel nitrate concentrations from three stream stations, and quantified the magnitude of diel amplitude and estimated the travel times between stations. The authors used a cross-correlation approach to conclude that instream processes controlled emergent diel signals, and were minimally driven by upstream inputs.

Overall, I enjoyed the paper, as it presents a means to interpret an essential ecohydrological question: which is more important, the physical or biological context, and

when do these abiotic/biotic controls matter most? It is also an interesting way to use spatially-explicit data, especially that which is emerging from the application of high-frequency sensors. I found the topic highly relevant, especially as high-frequency hydrochemistry paired with discharge is becoming more widely available, and questions about source pathways and mixing have become a topic of interest of the research community.

However, there were some points of confusion that I hope the authors can clarify in a revision. I have several main comments, and some minor ones mainly focusing on improving clarity of the manuscript, that I hope the authors find insightful.

Major Comments (1) While I am intrigued by the paper, one issue is that the authors overplayed the role of microbial processing. While this is generally assumed to be the case, this is still a "black box" situation with no microbial processing measured directly. I encourage the authors to take greater care in describing their findings and the assumptions of their interpretations, which as written are overly speculative.

(2) How were tributary inputs accounted for in the authors' approach (based on Figure 1 there were some small inputs in between monitoring stations)? Part of the difficulty in parsing apart nitrate removal/production processes is the fact that there is mixing happening from multiple landscape units, which are hydrologically mixed as tributaries meet, and it was not clear how this variability in inputs was accounted for in the authors approach.

(3) While the approach of using a time lag is compelling, I am curious if the authors had thought about the distributions of travel and reaction times in this study? The assumption of a mean travel time or reaction rate is to capture 'average' behavior and likely represents what is generally happening, but the use of a single value assumes that either transport or removal processes influencing what water/solutes make it to a point in the watershed network are occurring at a single rate. I am not encouraging the authors to use this approach, but it should likely be discussed as a potential limitation

of the study. Somewhat relatedly, why are there negative travel times in Figure 2, do you mean this to be the time lag?

(4) The authors could significantly shorten the discussion, as many of the processes mentioned were not directly measured and so the discussion does not need to be as nuanced as it is. Instead, presenting this as an open "call for the community" might be a more appropriate approach. Alternatively, one suggestion would be for the authors to develop a conceptual diagram of diel patterns in their watershed, indicating the open questions on the processes that the authors did not directly measure but infer as important instream drivers. Not only would this figure be useful for the community to visualize nitrate processing/transport in this system, but also likely hone the discussion around what is "known" and what is yet "unknown".

Minor Comments and Line-by-Line Suggestions

P1, Line 10: Change to "allow calculation"

P1, Line 15: Omit "suggested"

P2, Line 50: Please define insolation

P4, Section 2.2: Please describe in further detail how the s::can data were calibrated and turbidity-corrected.

P5, Section 2.3.1: Were the time lags / mean travel times estimated at the same intervals as the s::can data (i.e., did they also account for high/low Q, or are they averaged for a day)? Did you measure Q continuously at all three stations? Some additional clarity is needed here on time-scale and context for when travel times were estimated.

P11, Line 223: This sentence seems to come out of nowhere, I'd delete or expand on this idea before describing the Hensley & Cohen paper.

Figures & Tables

Generally, I thought the figure legends needed to have much greater detail.

For example, in the caption for Figure 2, r should be more clearly defined. I also wouldn't put the shading for the nominal travel time on the figure, as this looks like a regression or confidence interval.

In Figure 3, the letters should be defined in the figure legend and the confidence interval should be described.

Additionally, in the spirit of inclusivity, I encourage the authors to check that their figures are color-blind friendly.

---

## Author Comment (AC4) · 15 Feb 2021

**Author's response to comments by anonymous referee #4**

We would like to thank referee #4 for taking the time to carefully read our manuscript and for providing very constructive feedback. The helpful comments will improve the quality of our manuscript. Our responses to the individual comments are shown in blue below.

Summary The manuscript by Greiwe et al. describes a spatially-repeated sampling of diel variation in nitrate export along a reach in an intermediate watershed. The authors collected high-frequency diel nitrate concentrations from three stream stations, and quantified the magnitude of diel amplitude and estimated the travel times between stations. The authors used a cross-correlation approach to conclude that instream processes controlled emergent diel signals, and were minimally driven by upstream inputs. Overall, I enjoyed the paper, as it presents a means to interpret an essential ecohydrological question: which is more important, the physical or biological context, and when do these abiotic/biotic controls matter most? It is also an interesting way to use spatially-explicit data, especially that which is emerging from the application of highfrequency sensors. I found the topic highly relevant, especially as high-frequency hydrochemistry paired with discharge is becoming more widely available, and questions about source pathways and mixing have become a topic of interest of the research community.

However, there were some points of confusion that I hope the authors can clarify in a revision. I have several main comments, and some minor ones mainly focusing on improving clarity of the manuscript, that I hope the authors find insightful.

**Major Comments**

(1) While I am intrigued by the paper, one issue is that the authors overplayed the role of microbial processing. While this is generally assumed to be the case, this is still a "black box" situation with no microbial processing measured directly. I encourage the authors to take greater care in describing their findings and the assumptions of their interpretations, which as written are overly speculative.

> Reply: We agree that some of our claims are not satisfactorily backed up with data and will tone down our conclusions. We will also delete the reference to microbial processes in the title as suggested by some of the other referees.

(2) How were tributary inputs accounted for in the authors' approach (based on Figure 1 there were some small inputs in between monitoring stations)? Part of the difficulty in parsing apart nitrate removal/production processes is the fact that there is mixing happening from multiple landscape units, which are hydrologically mixed as tributaries meet, and it was not clear how this variability in inputs was accounted for in the authors approach.

> Reply: We agree that tributaries may alter diel stream nitrate patterns, and potentially even more critical, mass balances. There was only one surface input between S1 and S2. The influence of this tributary will be discussed in more detail in a revised manuscript. The streams visible outside the catchment boundaries in figure 1 in fact run parallel to the studied reach and enter the main stream downstream of S3.

(3) While the approach of using a time lag is compelling, I am curious if the authors had thought about the distributions of travel and reaction times in this study? The assumption of a mean travel time or reaction rate is to capture 'average' behavior and likely represents what is generally happening, but the use of a single value assumes that either transport or removal processes influencing what water/solutes make it to a point in the watershed network are occurring at a single rate. I am not encouraging the authors to use this approach, but it should likely be discussed as a

potential limitation of the study. Somewhat relatedly, why are there negative travel times in Figure 2, do you mean this to be the time lag?

>Reply: We agree that our approach is a simplification. In the real world, transport will be influenced by dispersion due to non-uniform cross-sectional flow profiles, hyporheic exchange and possibly other processes. These simplification will be discussed in a revised manuscript. Negative travel times in Figure 2, indeed, result from the time lag. We will make this clear and revise this figure according to the suggestion made by referee #1.

(4) The authors could significantly shorten the discussion, as many of the processes mentioned were not directly measured and so the discussion does not need to be as nuanced as it is. Instead, presenting this as an open "call for the community" might be a more appropriate approach. Alternatively, one suggestion would be for the authors to develop a conceptual diagram of diel patterns in their watershed, indicating the open questions on the processes that the authors did not directly measure but infer as important instream drivers. Not only would this figure be useful for the community to visualize nitrate processing/transport in this system, but also likely hone the discussion around what is "known" and what is yet "unknown".

>Reply: We agree that our discussion is too elaborate which was criticized by all referees. In a revised manuscript, we will shorten its speculative parts and stick closer to what we can really show with our data. Thank you for the suggestion with the conceptual diagram. This is a nice way to illustrate the key points of our considerations and we will sketch one in our revised manuscript..

**Minor Comments and Line-by-Line Suggestions**

P1, Line 10: Change to "allow calculation"

>Reply: Thank you, this will be adjusted.

P1, Line 15: Omit "suggested"

>Reply: Thank you, "suggested" will be deleted

P2, Line 50: Please define insolation

>Reply: We are referring to solar irradiance here and will clarify this in a revised manuscript.

P4, Section 2.2: Please describe in further detail how the s::can data were calibrated and turbidity-corrected.

>Reply: A detailed description of the calibration procedure will be added.

P5, Section 2.3.1: Were the time lags / mean travel times estimated at the same intervals as the s::can data (i.e., did they also account for high/low Q, or are they averaged for a day)? Did you measure Q continuously at all three stations? Some additional clarity is needed here on time-scale and context for when travel times were estimated.

>Reply: The cross-correlation approach yields one value per day. Although higher frequencies may be technically feasible using a floating window, lag estimates will be dominated by the lag between the strongest concentration excursions in the input signals. Higher frequencies will therefore not produce more information. Travel time estimates were based on daily means of water level recordings (15 min intervals, same as s::can probes) at the lowermost site S3 – no water level recordings are available from the other sites. We will improve clarity both in the text and also, as suggested by referee #1, by including a figure showing our raw data.

P11, Line 223: This sentence seems to come out of nowhere, I'd delete or expand on this idea before describing the Hensley & Cohen paper.

Reply: We will revise this paragraph to make it more understandable. This has also been noted by the other referees.

**Figures & Tables**

Generally, I thought the figure legends needed to have much greater detail. For example, in the caption for Figure 2, r should be more clearly defined. I also wouldn't put the shading for the nominal travel time on the figure, as this looks like a regression or confidence interval. In Figure 3, the letters should be defined in the figure legend and the confidence interval should be described. Additionally, in the spirit of inclusivity, I encourage the authors to check that their figures are color-blind friendly.

Reply: Thank your for pointing out the deficits in our figures. The issues with figure 2 and 3 have also been noted by the other referees and will be addressed correspondingly. We tried to act in the spirit of inclusivity by using the viridis color scheme, which was intentionally designed as color-blind friendly. However, we realize that contrasts between colors could be higher and will do our best to improve this.

---

## Author Response (AR1)

**Dear Dr. Battin,**

Please, find below a point-by-point response (blue) to the reviewers comments (black). We thoroughly revised our manuscript according to the suggestions of the four referees.

Major changes include

- to shift the focus from individual biochemical processes towards in-stream vs. transport control on diel nitrate patterns
- restructuring the manuscript so that we now use the identified clusters as a starting point for further analysis
- substantial shortening of the discussion and reduction of speculations.

We hope that we addressed all comments satisfactorily and that our revised manuscript now meets the requirements for publication.

Best regards,

Jan Greiwe

**Author's response to comments by anonymous referee #1**

**Summary**

This paper examines patterns and sources of diel variation in stream NO3 concentration along a lowland river in Germany. The authors show that diel patterns of stream NO3 concentration vary over the growing season, yet most days show similar diurnal oscillations. Further, by combining different statistical techniques, the authors convincingly show that diel patterns are mostly driven by in-stream processes. Finally, the authors analyze diel and seasonal patterns of several environmental variables to discuss which in-stream process is driving diel NO3 cycles.

**General comments**

This paper makes a significant contribution to watershed and stream ecology through its assessment of patterns and controls of diel variation in stream NO3 concentration. However, I have some major issues that need to be addressed. All comments are made in the spirit of increasing the potential impact of this interesting research.

1. While most of the findings presented in the paper are original and compelling, the conclusions raised from them are sometimes speculative and inaccurate. For instance, the authors concluded that "the magnitude of microbial NO3 processing may be large compared to plant uptake", but they did not measure any in-stream process (GPP, denitrification, nitrification) nor NO3 uptake rates. Hence, it is impossible to know, based on their data and results, which in-stream process was contributing the most to NO3 uptake rates over the study period. Similarly, they stated that "diel patterns in NO3 concentration suggest the importance of microbial pathways for in-stream processing", but the 70% of diel patterns seem to be driven by photoautotrophic uptake (not microbial pathways). My suggestion is to focus the objectives and conclusions on the compelling results and only speculate about the relative importance of different in-stream processes in the discussion.

Reply: In concordance with the suggestions by referee #1, we shifted the focus of the paper towards in-stream vs. transport control on diel nitrate patterns and accordingly changed the title to "*Diel patterns in NO3*" concentration produced by in-stream processes".

We also revised the conclusions accordingly (l. 318-331):

"In a 5.1 km stream reach of the river Elz in Southwest Germany we identified diel patterns in stream NO3- concentration, differentiated between in-stream and transport control, and analyzed how patterns were related to environmental conditions and potential drivers. We found a set of six clusters representing different characteristic diel  $NO_3^-$  patterns. Relatively small temporal shifts between adjacent monitoring sites indicated that  $NO_3^-$  concentration patterns were predominantly formed by in-stream processes and not by a transport of upstream  $NO_3^{-1}$  inputs. Most patterns were characterized by a pre-dawn maximum and an afternoon minimum of varying intensity, and mostly the change rate of  $NO_3^-$  concentration was negatively correlated with global irradiance. We therefore conclude that these patterns were primarily produced by photoautotrophic  $NO_3^-$  uptake. However, we also found indications that other biochemical processes like nitrification and denitrification contributed to the formation of  $NO_3^{-}$  patterns. In depth interpretation and eventually quantification of process rates would require spatially distributed high frequency information on stream metabolism, e.g. dissolved oxygen concentrations, and on different N species, most importantly NH4+. Nevertheless, our analysis suggests that particular combinations of different in-stream processes may generate distinct diel NO3- patterns. A seasonal shift in patterns may then indicate shifts in the relative importance of the underlying processes. The clustering method used in this study proved useful for making the data set accessible for this kind of analysis and may be used as a blueprint for the analysis of other stream solutes."

2. I missed some results regarding lateral inputs. In the discussion, the authors mentioned that lateral inputs may not affect diel NO3 patterns because they did not observe diel variations in discharge. While I agree with this statement, lateral inputs should be included in the hypothesis, methods and results (see Flewelling et al. 2014 or Lupon et al. 2016). Also, the authors mentioned that there was a tributary entering to the upstream reach. Does the tributary show diel variation in NO3 concentration? How this may influence stream NO3 concentration in S2?

Reply: We added a section in the discussion that explicitly deals with lateral inputs (l. 254-273):

"Diel NO3- patterns may also be influenced by lateral inputs, including tributaries and groundwater interaction. The only surface tributary within the studied stream reach was between S1 and S2. It was initially considered negligible and therefore not accounted for. However, snap shot sampling on a hot day during low flow conditions revealed nitrate concentration to be twice as high as in the main stream. It is also possible that groundwater influx influenced  $NO_3$  concentration at the monitoring sites. In fact,  $NO_3$  levels in groundwater were higher than in stream water in the proximity of the upper reach and lower than in stream water along the lower reach (Fig. S3). Although the overall flow direction of groundwater was parallel to the stream, groundwater inputs might explain the increase in average  $NO_3^-$  concentration from S1 to S2 and subsequent decrease from S2 to S3 (Fig. S2). Previous research identified diffuse groundwater inputs as a considerable challenge for determining mass balances using paired high-frequency probes (Kunz et al., 2017). We were unable to separate the effects of groundwater inputs from a potential effect of increased  $NO_3$ removal in the lower reach due the revitalization measures. Although lateral inputs may have affected average  $NO_3^-$  levels, their influence on diel  $NO_3^$ patterns was only marginal. In the upper reach, which received the tributary, diel  $NO_3^{-1}$ patterns were mostly longitudinally stable, except for the deployment in September (Fig. 3). We therefore consider the influence of the tributary to be limited. Riparian groundwater interaction induced by evapotranspiration was suggested by Aubert and Breuer (2016) to explain a seasonal shift in diel  $NO_3^-$  patterns. Flewelling et al. (2014) showed that diel fluctuations in groundwater level and stream flow induced by evapotranspiration may be sufficient to produce measurable diel patterns in stream  $NO_3^-$  concentration. Groundwater inputs may not only directly affect  $NO_3^-$  concentrations but also alter stream chemistry, e.g., by introducing labile organic carbon which promotes heterotrophic processes (Lupon et al., 2020). In the present study, however, diel water level fluctuations were usually minimal so that we generally have little evidence for diel variability in groundwater influx."

3. I was confused by some of the approaches used. For instance, what is the point of the mass balance? It has many uncertainties (e.g. groundwater, tributaries) and the results derived from it are difficult to interpret. My suggestion is to delete this whole section. Instead, I will focus on analyzing (i) if all sites showed similar seasonal patterns in diel NO3 variation (i) if the effect of longitudinal propagation differed across clusters; (iii) if there was a lag time between diel patterns of drivers and stream NO3 concentration (see my specific comments for more info on this regard).

Reply: It is true that the mass balances have many uncertainties. We therefore removed them from the manuscript but, for the interested reader, show the distribution of concentrations at the monitoring sites in the supplementary material (Fig. S2). We addressed the suggestions made by referee #1 as follows.

 Seasonal patterns at monitoring sites are compared in the new Fig. 3 (see below) and in the text (l. 177-180): "In terms of cluster occurrence, a largely similar seasonal pattern was apparent at all monitoring sites, despite different numbers of recorded days. Cluster A dominated in May and again in October and was replaced by cluster B during the summer months from June to September. Both clusters usually formed continuous blocks of several days. Cluster C occurred occasionally throughout the season but preferentially in early summer, while cluster D and E mainly occurred in fall."

- (ii) We tested if longitudinal propagation differed among clusters by including clusters in the revised Fig. 2 (Fig. 4 in the revised manuscript, see below). This is described in 1. 196-199: "In the lower reach, lags formed an evenly distributed point cloud. Within this cloud, Cluster D, E, and F only appear at above median flows. In the upper reach, time lags were concentrated towards the extremes, i.e. either close to zero or close to travel time estimates. Days with below median stream flow were mainly assigned to cluster B and those above median stream flow to cluster A."
- (iii) Time lags between potential drivers and nitrate concentration is apparent in the different clusters as timing of drivers was more or less constant throughout the year. We think that this topic is sufficiently addressed by the corresponding correlations and Fig. 6 (see below) (1. 215-220): "In addition to different environmental conditions, we identified different relationships with potential drivers of diel cycles among clusters (Fig. 6). The correlation of  $\delta C_{diel}$  and S was positive in cluster D, negative in clusters A and C, and strongly negative in cluster B. Moderate correlations of  $\delta C_{diel}$  with h were found in cluster C (negative) and cluster E (positive). Correlations of  $\delta C_{diel}$  with h were weak and difference among clusters were less pronounced than with S and T. The relationship of  $C_{obs}$  and h was very variable and included both strongly positive and negative correlations. However, strong overlapping of boxplots in Fig. 6c and Fig. 6d indicated that variability within clusters was higher than among cluster."

4. The discussion is a little bit puzzling. My suggestion is to delete all sub-headings and focus on how different sources shape stream NO3 concentration. You can start with a paragraph discarding longitudinal propagation and lateral inputs as factors causing diel NO3 patterns. Then, move to the most obvious process: photoautotrophic uptake (clusters A-B) and how it varies over time depending on light, temperature, discharge. Finally, you can suggest potential explanations for the other clusters: denitrification (cluster C), nitrification (cluster D), storm flow (cluster F).

Reply: The discussion has been shortened and is now devided into 5 sections: "General patterns" (l. 232-241), "In-stream vs. transport control" (l. 242-253), "Lateral inputs" (l. 254-273), "Interpretation of diel patterns" (l. 274-316), "Conclusions" (l. 317-331). In the "Interpretation of diel patterns" section, we substantially reduced speculations and implemented the suggestions made above. It now reads (l. 274-316):

[revised manuscript text omitted]

5. While I like the figures, most of them (and their captions) need some improvements (see my specific comments). Also, I missed a figure showing the raw data (i.e. diel patterns of NO3, discharge, light and temperature over the whole study period). This figure is key to understand some of the points discussed (e.g. no diel variation in discharge); and it will be very helpful to the readers.

Reply: The figures have been revised and an additional figure (Fig. 3) showing the raw data was added.

---

## Author Response (AR2)

**Response to referee #1**

We thank referee #1 for once again for the helpful advise and hope that we have satisfactorily address all concerns.

**Recommendation to the editor**

**Suggestions for revision or reasons for rejection (will be published if the paper is accepted for final publication)**

This is my second review of this paper. The authors have properly addressed most of my original concerns. My only major comment is to better organize the section on interpreting the clusters, preferably with a visualization. I think this would help with the interpretability of the hypothesized controls of the clusters and would make a powerful contribution.
A figure (Figure 7) was added that leads through the interpretation of clusters in the discussion section.
Other minor comments are below.

Line 28: End the abstract with a broad conclusion statement.
Done (l. 19-20)
Line 145: How did the water authority measure discharge? At least list the method.
Done (l. 92)
Line 287: It still isn't clear to me where the 95% CI come from. A model (time-series, regression, etc.) must have been used to get these values, and I don't see anything about a model in the methods.
The intervals show the spread of the original data. No model is involved. In order to avoid misunderstanding, we decided to show the individual recordings instead of shaded areas in a modified version of figure 2.
Line 325/ Figure 4: This is still challenging to interpret. If the points were close to the shaded area does that mean that transport dictated the difference in NO3 between sites. In other words, when the values are less than the shaded area, differences in NO3 between sites are caused by in-stream processes. Please add a line in the caption that describes the main conclusion to take away from this graph.
D We added the sentence *"Points falling below the shaded areas indicate in-stream control on diel NO3- patterns, whereas points within the range of travel time estimates suggest transport control"* to the figure caption.
Also, the authors have not addressed my comment about how its possible to have a negative travel time.
Travel times cannot not be negative, but time lags produced by cross-correlation can. We removed the misleading axis label and added information to the figure caption.
Line 563: Yes, if GW N inputs are constant, they shouldn't affect the signal (shape of the diel curve). Although, GW N inputs would mess with your interpretation of transport vs. in-stream processes. I would talk about that here.
We understand diffuse groundwater inputs as features of a longer stream reach (as opposed to point sources). In our differentiation between in-stream and transport control, we consider them as in-stream condition. We clarified this in the introduction (l 53-55.):

*"In this sense, we use the term 'in-stream' to refer to average properties of a reasonably long stream reach and its immediate surroundings including biochemical conditions in the stream and in the hyporheic zone as well as diffuse groundwater interaction."*

Line 564: What does longitudinally stable mean? It implies stability over space, say between sensors, but that doesn't fit with the context of this sentence. Do you mean stabile in time, e.g., a similar pattern over multiple days? Please describe.

**The concept of longitudinal stability is introduced in the methods section (l. 116-118). We further clarified the sentence in question. It now reads:** *"Yet, the influence on patterns in the main stream must have been small as the patterns were usually longitudinally stable, i.e. the same upstream (S1) and downstream (S2) of the tributary."* **(l.259-260)**

Line 585-589: I would remove this reference. If there is no riparian vegetation in your stream this is irrelevant.

**Done**

Line 603: How would diel temperature variation affect denitrification and not nitrification? This correlation should be described more fully.

**We reworked our interpretation of clusters C to E including our hypothetical explanations for the clusters. The section now reads:**

*" Patterns with a midday maximum such as those observed in cluster C are hard to explain by photoautotrophic assimilation 315 alone in systems without intense seasonal shading by riparian vegetation (as opposed to e.g. Rusjan and Mikoš (2010)). Figure 7 (a, b, c, f) shows that $NO_3$- peak time gradually shifts towards midday when photoautotrophic assimilation decreases and the relative importance of a $NO_3$- depleting process negatively related with temperature increases. This suggests that either denitrification or heterotrophic assimilation or both are promoted by stream temperature and drive the shape of the signal. Diel variability has been observed both in denitrification (Christensen et al., 1990; Harrison et al., 2005; Cohen et al., 2012) and 320 heterotrophic respiration (Hotchkiss and Hall, JR., 2014) which is closely linked to heterotrophic $NO_3$- assimilation. However, peak $NO_3$- depletion occurs in the afternoon, when oxygen levels are expected to be elevated and unfavorable for anaerobe denitrification (Rysgaard et al., 1994). In addition, it is not clear how denitrification in the lower anoxic sediments could be promoted by temperature without simultaneously increasing nitrification in the upper sediment layers. However, it seems possible that the driving force of cluster C was not temperature but another process with similar diel pattern. Exudation of 325 algal photosynthate rich in labile organic carbon (Kaplan and Bott, 1982) may have a similar diel course and stimulate assimilation and denitrification by heterotrophs but not nitrification by autotrophs. Under such conditions, heterotrophic assimilation or denitrification or both may drive diel $NO_3$- fluctuation in cluster C.*

*In literature, diel patterns with a $NO_3$- peak in the afternoon (cluster D) have been attributed to intense evapotranspiration (Aubert and Breuer, 2016; Flewelling et al., 2014; Lupon et al., 2016a). In the present study, evapotranspiration was not 330 measured, however, it did not produce systematic diel fluctuations in water level and the latter were not correlated with diel $NO_3$- signals. Such patterns could also not be reproduced by overlaying rates of light-dependent and temperature-dependent processes (Fig. 7). An explanation for cluster D is therefore still warranted.*

*Diel patterns with a midday low (cluster E) could be the result of low photoautotrophic assimilation and a temperature-dependent $NO_3$- producing processes like nitrification (Fig. 7g). Diel variability in nitrification is well documented (Warwick, 335 1986; Laursen and Seitzinger, 2004; Dunn et al., 2012) and it seems principally plausible that temperature promotes*

*nitrification without influencing denitrification in deeper anoxic sediment layers. Another reason for independence of nitrification and denitrification may be limitation of heterotrophic denitrification in absence of an organic carbon source. " (l.315-338)*

Also, what about the possibility of diel changes in heterotrophic respiration (see: Hotchkiss and Hall 2014, https://doi.org/10.4319/lo.2014.59.3.0798) . How might that describe some of the clusters?

**Thank your for this hint. Heterotrophic respiration is now part of our hypotheses and is considered to potentially explain cluster C. S. above**
Line 608-609: But above you seem definitive that diel changes in lateral inputs are negligible. Which one is it?

**Based on water level measurements, potential diel variability in hydraulic gradient between groundwater and stream water was minimal. This is now stated in lines 329-333:**
*" In literature, diel patterns with a $NO_3^-$ peak in the afternoon (cluster D) have been attributed to intense evapotranspiration (Aubert and Breuer, 2016; Flewelling et al., 2014; Lupon et al., 2016a). In the present study, evapotranspiration was not measured, however, it did not produce systematic diel fluctuations in water level and the latter were not correlated with diel $NO_3^-$ signals. Such patterns could also not be reproduced by overlaying rates of light-dependent and temperature-dependent processes (Fig. 7). A satisfactory explanation for cluster D is therefore still needed."*

Lines 585-615: Thank you for toning down the language about the causes of the diel patterns. However, this section seems to have lost its teeth. You are definitely on the right track in using nitrification and denitrification to explain the clusters that aren't closely related with autotrophic based assimilation, but the current explanation is disjointed and could use some organization.
I suggest creating a conceptual diagram/figure with hypotheses about the controls on each cluster, and using this figure to write about each cluster in this section. Frame it as hypotheses that readers can use in their research to pursue controls on diel trends that they see. I think this would be very powerful and useful for folks that use diel data. You are on the right track, I just think a little organization and a visualization would make what you are describing more obvious and insightful.

**A corresponding figure and introductory text was added (l. 281-295):**
*" In the following, we therefore aim to interpret our findings based on in-stream processes. In terms of biochemical processes, diel $NO_3^-$ variability depends on the time-varying balance of $NO_3^-$ removal (via assimilation by both heterotrophs and autotrophs as well as denitrification) and $NO_3^-$ production (via mineralization and subsequent nitrification). We do not regard our interpretations on the controls of the observed patterns complete but as hypotheses for further research on diel dolute patterns to build upon. Considering the idea of multiple superposed biochemical processes as a starting point, some assumptions can be made on the diel course of the processes mentioned above. Photoautotrophic assimilation depends on the light availability and can be conceptualized as a function of solar irradiance. In contrast, the degree of diel variability in nitrification, denitrification, and heterotrophic assimilation is less clear. However, we generally assume that the rate of microbial metabolism (besides other influences) increases with temperature. Figure 7 schematically illustrates the diel concentration signals resulting from overlaying the diel courses of light-dependent photoautotrophic ($r_{aa}$) assimilation and a complementary temperature-dependent processing rate ($r_{comp}$). The latter represents the combined net effect of heterotrophic assimilation, nitrification and denitrification. Particularly,*

*we consider different levels of light intensity (columns in Fig. 7) and different types of relationship with temperature (rows in Fig. 7), including positive and negative correlation with temperature as well as constant $r_{comp}$. The shapes of $r_{aa}$ and $r_{comp}$ reflect means over all measurements of global irradiance and water temperature, respectively, during the course of the present study. ”*

[Figure]

Figure 1: *Schematic representation of diel courses of assimilation rate ($r_{aa.}$) and a complementary processing rate ($r_{comp}$) required to produce equilibrium conditions. Black lines show the resulting change rate ($r_{res}$) and concentration (C). While $r_{aa}$ is considered a function of global irradiance (columns represent different levels of irradiation intensity), $r_{comp.}$ is conceptualized as a function of stream temperature (rows represent a negative (a-c), no relationship (d-f) and a positive relationship (g-i)).*

**Response to referee #2**

We thank referee #2 once again for the helpful advise and hope that we have satisfactorily address all concerns.

**Recommendation to the editor**

**Suggestions for revision or reasons for rejection (will be published if the paper is accepted for final publication)**

I very much enjoyed reading this revised manuscript. The authors did an excellent job of responding to reviewer questions and concerns, and I find the paper much improved.

I still have some (very) minor comments and technical notes that I am sure the authors can easily solve.

Minor comments

1. I suggest a small **reordering in the methods and results**. In section 2.3.1., I would begin with seasonal patterns of stream nitrate concentrations and then move to diel patterns. (i.e. move lines 109-113 earlier in this section). Same in the results (i.e. move lines 167-176 at the beginning of section 3.1).
*The section have been reordered accordingly.*

2. Looking at Fig. 5, I wonder if it is possible to add **some test (i.e. ANOVA)** that support your results. This shouldn't be difficult to do, and it might support your rational.
*An ANOVA and post hoc Tukey HSD test were performed on the cluster data. Corresponding changes were made in the methods section (l. 144-146):*
*" Differences between clusters were statistically assessed using analysis of variance (ANOVA) and Tukey honestly significant difference (HSD) tests as implemented in the 'stats' R-package (R Core Team, 2019)."*
*, the results (l. 215-226),*
*" We found clear differences in the distribution of daily means of environmental parameters among clusters (Fig. 5). The following characterization of the clusters refers to significant differences ($p<0.05$) according to Tukey HSD test applied to an ANOVA on the cluster data. Cluster A presented overall lowest $NO_3^-$ concentrations (median 4.36 mg $L^{-1}$) which differed from those during cluster B (median 4.87 mg $L^{-1}$) and C (median 4.88 mg $L^{-1}$). Cluster A also showed the lowest water temperature (median 14.1 °C) and elevated water levels (median 41.9 cm) compared to cluster B and C. Cluster B was characterized by the highest global irradiance (median 825.0 W/m²), highest water temperature (21.7 °C) and lowest water levels (median 21.2 cm). Disregarding cluster F with only 3 data points, the difference in water temperature was significant for all remaining clusters. Global irradiance in cluster B differed from all clusters but C. Water level differed with all clusters but C and E. Cluster C occurred during very similar conditions as cluster B and only differed from cluster B in terms of water temperature (median 16.4 °C). The two clusters D and E were characterized by lower global irradiance than cluster*

*B and C and did not differ from one another. Cluster F consisted of only 3 days, but all of these represented water levels (median 77.2 cm) hardly ever observed in the remaining clusters.*"
**and in figure 2:**

"

[Figure]

*Figure 2: Environmental conditions during occurrence of clusters. The panels show daily average $NO_3^-$ concentration (a), daily maximum of global irradiance (b), daily average water temperature (c), and daily average water level (d). Lowercase letters above boxplots were assigned to groups that do not differ significantly according to analyses of variance (ANOVA) and Tukey tests.*"

3. Finally, I still have some minor concerns about the discussion.

First, **the section "lateral inputs" is not needed**. The first paragraph of this section can easily go to "General patterns", as the authors talk about how lateral inputs affect longitudinal patterns of stream nitrate concentrations at seasonal scale.
**The first paragraph was moved to the "general patterns" section (l.255-268)**

Similarly, the second paragraph links nicely to line 308, where the authors say that cluster D might be caused by intense evapotranspiration.

**The second paragraph was partly deleted due to redundancy with the interpretation of cluster D.**

Similarly, I think the section "in-stream vs transport" can easily fit in "interpretation of diel patterns".

**The section on "in-stream vs transport" now forms the first paragraph of the "interpretation of diel patterns" section (l. 270-280).**

The rational on how evapotranspiration influences diel patterns is confusing. The authors said that **intense evapotranspiration** can cause cluster D. However, the authors also said that there were no diel patterns of water level, which goes against the previous hypothesis. Please, clarify it.

**It is true that diel changes in water level were very minimal. We revised the paragraph on cluster D. We now refer to cases where similar observations were explained by evapotranspiration, but make clear that this was very unlikely in our study (l. 329-333):**

*" In literature, diel patterns with a $NO_3^-$ peak in the afternoon (cluster D) have been attributed to intense evapotranspiration (Aubert and Breuer, 2016; Flewelling et al., 2014; Lupon et al., 2016a). In the present study, evapotranspiration was not measured, however, it did not produce systematic diel fluctuations in water level and the latter were not correlated with diel $NO_3^-$ signals. Such patterns could also not be reproduced by overlaying rates of light-dependent and temperature-dependent processes (Fig. 7). A satisfactory explanation for cluster D is therefore still needed. "*

I did not follow **why the relation between Cdiel and water temperature suggest that nitrification denitrification may be the underlying processes.** Please, can you develop a little bit more this rational?

**We added a figure (Fig. 7) to illustrate how we imagine the superposition of different processes (as requested by referee 1) and revised the interpretation of the individual clusters (l. 303-338):**

[Figure]

*Figure 3: Schematic representation of diel courses of assimilation rate ($r_{aa.}$) and a complementary processing rate ($r_{comp}$) required to produce equilibrium conditions. Black lines show the resulting change rate ($r_{res}$) and concentration (C). While $r_{aa}$ is considered a function of global irradiance (columns represent different levels of irradiation intensity), $r_{comp.}$ is conceptualized as a function of stream temperature (rows represent a negative (a-c), no relationship (d-f) and a positive relationship (g-i)).*

**Technical notes**

Title: "stream nitrate concentration"?
**Done**
Ln 9. This sentence is quite long. I suggest to start a new one on "We performed a k-means cluster analyses to ....".
**Done**
Ln 15. Perhaps add "Results from cluster analyses show that at least 70% …"
**Done**
Ln 16: "or physical (lateral inputs) processes"
**Done**
Ln 2. This sentence seems quite off here. I suggest to either move it at the beginning of the introduction or to delete it and simply say "Among the different nutrients, nitrate is of special interest …"
**Done**
Ln 42. Add here also the hypothesis that diel signals might also be due to diel variations in groundwater NO3- inputs due to evapotranspiration.
**Done**
Ln 50. Move the ref at the end of the sentence.

**Done**

Ln 61. "Al our site, "

**Done**

Ln 66. "gravel bed,"

**Done**

Ln 88. "in the latter dilution measurement,"

**Done**

Ln 92. Perhaps is better to indicate which patterns of NO3- concentrations you are referring to. I suggest to change the heading to "Identification of diel patterns in stream NO3- concentration"

**Done**

Ln 99. Remove "e.g."

**Done**

Ln 100. You did a great job here; this reads much clearer now! However, I would perhaps clarify that "diel portion of the solute concentration signal" means "Residuals concentrations", i.e. add residuals concentrations in parenthesis after "concentration signal".

**Instead we replaced "Residual" with "diel" in the following sentence to overall avoid the term "residual" which had been suggested in the first revision cycle.**

Ln 124. Clarify that you did this exercise for both reaches and all clusters. Also, can you add the number of days for each reach separately? If you do so, data would match perfectly with Fig 4.

**Done**

Lnn138, "Particularly, we assessed daily means of nitrate concentration, water level and water temperature"

**Done**

Ln 155. Add a comma after "(n=119)"

**Done**

Lm 166. "stream nitrate concentration"

**Done**

Ln 172. This rho is negative. So either the symbol is wrong, or the relation between water temperature and daily average nitrate concentration is negative.

**Typo corrected**

Ln 183. "differed between stream reaches and among clusters"

**Done**

Ln 194. "In both reaches, the time lags roughly ranged between zero and travel time estimates"

**Done**

Fig 4. Nitpicking, but perhaps it is better to show first the upper reach.

**Done**

Ln 232: "In our data, we found patterns in stream nitrate concentration"

**Done**

Ln 238. "On diel scale, "

**Done**

Fig 6. Nitpicking, but can you highlight the zero line to make it easier for the reader?

**Done**

Ln 270-274. This rational does not make sense in the current version of the discussion. I suggest deletion.

**Done**

Ln 281. ", when cluster B prevailed"

**Done**
Ln 319. "Germany,"
**Done**